# Learning Generalized Policy Automata for Relational Stochastic Shortest Path Problems

**Rushang Karia, Rashmeet Kaur Nayyar, Siddharth Srivastava**
School of Computing and Augmented Intelligence
Arizona State University
Tempe, AZ 85281
U.S.A.
{Rushang.Karia,rmnayyar,siddharths}@asu.edu

## Abstract

Several goal-oriented problems in the real-world can be naturally expressed as Stochastic Shortest Path problems (SSPs). However, the computational complexity of solving SSPs makes finding solutions to even moderately sized problems intractable. State-of-the-art SSP solvers are unable to learn generalized solutions or policies that would solve multiple problem instances with different object names and/or quantities. This paper presents an approach for learning *Generalized Policy Automata* (GPAs): non-deterministic partial policies that can be used to catalyze the solution process. GPAs are learned using relational, feature-based abstractions, which makes them applicable on broad classes of related problems with different object names and quantities. Theoretical analysis of this approach shows that it guarantees completeness and hierarchical optimality. Empirical analysis shows that this approach effectively learns broadly applicable policy knowledge in a few-shot fashion and significantly outperforms state-of-the-art SSP solvers on test problems whose object counts are far greater than those used during training.

## 1 Introduction

Goal-oriented Markov Decision Processes (MDPs) expressed as Stochastic Shortest Path problems (SSPs) have been the subject of active research since they provide a convenient framework for modeling the uncertainty in action execution that often arises in the real-world. Recently, research in deep learning has demonstrated success in solving goal-oriented MDPs using image-based state representations (Tamar et al., 2016; Groshev et al., 2018). However, such methods require significant human-engineering effort in finding transformations like grayscale conversion, etc., to yield representations that facilitate learning. Many practical problems however, are more intuitively expressed using relational representations and have been widely studied in the literature.

As an example, consider a planetary rover whose mission is to collect all rocks of interest from a planet's surface and deliver them to the base for analysis. Such a problem objective is not easily described in an image-based representation (e.g., visibility is affected by line of sight) but can be easily described using a relational description language such as first-order logic. Finding suitable image-based representations for such problems would be counter-productive and difficult. Furthermore, image-based deep learning methods often require large amounts of training data and/or are unable to provide guarantees of completeness and/or convergence.

Many real-world problems such as the rover example above can be readily expressed as SSPs using symbolic descriptions that can be solved in polynomial time in terms of the state space. SSP algorithms use a combination of pruning strategies (e.g., heuristics (Hansen and Zilberstein, 2001)) that can eliminate large parts of the search space from consideration thereby reducing the computational effort

expended. In spite of such optimizations, a major hurdle is the "curse-of-dimensionality" since the state spaces grow exponentially as the total number of objects increases. Existing SSP solvers would have difficulty scaling to rover problems with many locations and/or rocks. The pruning strategies employed by these SSP solvers do not scale well because they do not use knowledge that could have been exploited from solving similar problems. One solution to this problem is to compute a simple *generalized policy*: move the rover to the closest available location with an interesting rock, try loading the rock until it succeeds, navigate back to the base, unload it, and reiterate this process until all interesting rocks are at the base. This generalized policy can be used to solve any rover instance with larger numbers of locations and rocks sharing a similar goal objective.

Related work in Generalized Planning addresses the problem of computing generalized policies by learning reliable controllers for broad classes of problems (Srivastava et al., 2008; Bonet et al., 2009; Aguas et al., 2016). Recently, Deep Learning based approaches have been shown to successfully learn such policies (Toyer et al., 2018; Garg et al., 2020; Karia and Srivastava, 2022). A key limitation of such methods is the lack of interpretability and theoretical guarantees of the learned policy. In this paper, we show that interpretable, generalized policies can be learned with guarantees of completeness and hierarchical optimality using solutions of very few, *small* problems with few objects.

The primary contribution of this paper is a novel approach for few-shot learning of *Generalized Policy Automata* (GPAs) using solutions of SSP instances with small object counts. GPAs are non-deterministic partial policies that represent generalized knowledge that can be applied to problems with different object names and larger object counts. This process uses logical feature-based abstractions to lift instance-specific information like object names and counts while preserving the relationships between objects in a way that can be used to express generalized knowledge. GPAs learned using our approach can be used to *accelerate any model-based SSP solver* by pruning out large sets of actions in different, related, but larger SSPs with many objects. We prove that our approach is complete and guarantees hierarchical optimality. Empirical analysis on a range of well-known benchmark domains shows that our approach few-shot learns GPAs using as few as three training problem instances and convincingly outperforms existing state-of-the-art SSP solvers and does so without compromising the quality of the solutions found.

The rest of this paper is organized as follows: The next section provides the necessary background. Sec. 3 describes our approach for using example policies in conjunction with abstractions to learn GPAs and using them for solving SSPs. We present our experimental setup and discuss obtained results in Sec. 4. Sec. 5 provides a description of related work in the area. Finally, Sec. 6 states the conclusions that we draw upon from this work followed by a brief description of future work.

## 2   Background

Our problem setting considers SSPs expressed in a symbolic description language such as the Probabilistic Planning Domain Definition Language (PPDDL) (Younes et al., 2005). Let $\mathcal{D} = \langle \mathcal{P}, \mathcal{A} \rangle$ be a problem *domain* where $\mathcal{P}$ and $\mathcal{A}$ are finite sets of predicates and parameterized actions. Object *types*, such as those used in PPDDL, can be equivalently represented using unary predicates. A relational SSP problem instance for a domain $\mathcal{D}$ with a goal formula $g$ over $\mathcal{P}$ and a finite set of objects $O$ is defined as a tuple $P = \langle O, S, A, T, C, s_0, g \rangle$. A fact is the instantiation of a predicate $p \in \mathcal{P}$ with the appropriate number of objects from $O$. A state $s$ is a set of true facts and the state space $S$ is defined as all possible sets of true facts derived using $\mathcal{D}$ and $O$. Similarly, the action space $A$ is instantiated using $\mathcal{A}$ and $O$. $T : S \times A \times S \to [0, 1]$ is the transition function and $C : S \times A \times S \to \mathbb{R}^+$ is the cost function. An entry $t(s, a, s') \in T$ defines the probability of executing action $a \in A$ in a state $s \in S$ and ending up in a state $s' \in S$, and $c(s, a, s') \in C$ indicates the cost incurred while doing so. Naturally, $\sum_{s'} t(s, a, s') = 1$ for any $s \in S$ and $a \in A$. Note that $a$ refers to the instantiated action $a(o_1, \ldots, o_n)$, where $o_1, \ldots, o_n \in O$ are the action *parameters*. We omit the parameters when it is clear from context. $s_0 \in S$ is a *known* initial state. A goal state $s_g$ is a state such that $s_g \models g$. For any action $a \in A$ and any goal state $s_g \in S$, $c(s_g, a, s_g) = 0$ and $t(s_g, a, s_g) = 1$. Additionally, termination (reaching a state $s$ such that $s \models g$) in an SSP is inevitable making the length of the horizon unknown but finite (Bertsekas and Tsitsiklis, 1996).

**Running example:** The planetary rover example can be expressed using a domain that consists of predicates *connected*$(l_x, l_y)$, *interesting*$(r_x)$, *rover-at*$(l_x)$, *in-rover*$(r_x)$, *rock-at*$(r_x, l_x)$, and parameterized actions *load*$(r_x, l_x)$, *unload*$(r_x, l_x)$, and *move*$(l_x, l_y)$. Object types can be denoted using

unary predicates $location(l_x)$ and $rock(r_x)$. $l_x$, $l_y$, and $r_x$ are *parameters* that can be instantiated with different locations and rocks thus allowing for an easy way to express different problems. Action dynamics are described using closed-form probability distributions (e.g., loading a rock could be modeled so that the rover picks up the rock with a probability of 0.8) and this forms the transition function. A simplified SSP problem that ignores connectivity and consists of two locations, a base location, and two rocks can be described using a set of objects $O = \{l_1, l_2, l_{base}, r_1, r_2\}$. A state in this SSP $s_{eg}$ that describes the situation where $r_2$ is being carried by the rover and $r_1$ is at $l_2$ can be written as $s_{eg} = \{location(l_1), location(l_2), location(l_{base}), rock(r_1), rock(r_2), in\text{-}rover(r_2), rock\text{-}at(r_1, l_2)\}$. The goal of delivering all the rocks to the base can be expressed as $\forall x \; rock(x) \implies rock\text{-}at(x, l_{base})$. Executing any action can be assumed to expend some fuel and as a result, the objective is to deliver all the rocks to the base in a way that minimizes the total fuel expended.

A solution to an SSP is a deterministic policy $\pi : S \to A$, that is a mapping from states to actions. A *proper policy* is one that is well-defined for all states. A *complete proper policy* is one for which termination is guaranteed from all possible states. By definition, SSPs must have at least one complete proper policy (Bertsekas and Tsitsiklis, 1996). This can be overly limiting in practice since such a formulation does not model dead end states: states from which the goal is reachable with probability 0. A weaker formulation of an SSP stipulates that the goal must be reachable with a probability of 1 from $s_0$ i.e. whose solution is a *partial proper policy* that is well-defined for every state reachable from $s_0$. As such, we focus on a broader class of relaxed SSPs called Generalized SSPs (GSSPs) (Kolobov et al., 2012) that only require the existence of at least one partial proper policy. Henceforth, we use the term SSPs to refer to GSSPs and focus only on partial proper policies.

The value of a state $s$ when using a policy $\pi$ is the expected cost of executing $\pi(s)$ when starting in $s$ and following $\pi$ thereafter: $V^\pi(s) = \sum_{s' \in S} t(s, \pi(s), s')[c(s, \pi(s), s') + V^\pi(s')]$ (Sutton and Barto, 1998). Naturally, $V^\pi(s_0) = \infty$ if $\pi$ is not a partial proper policy. $V$ is known as the value function. $V^*$ is the optimal value function and can be described using the *Bellman optimality equation*: $V^*(s) = \min_{a \in A} \sum_{s' \in S} t(s, a, s')[c(s, a, s') + V^*(s')]$

In this paper, we focus on optimality w.r.t. a given initial state. The optimal policy $\pi^*$ for an SSP $P$ w.r.t. $s_0$ is a policy that is better than or equal to all other policies i.e. $V^{\pi^*}(s_0) \le V^\pi(s_0)$ for any policy $\pi$. SSP solvers iteratively apply the Bellman optimality equation starting from $s_0$ to compute a policy, and under certain conditions, have been proved to converge to a policy that is $\epsilon$-consistent with $\pi^*$ (Hansen and Zilberstein, 2001; Bonet and Geffner, 2003).

Let $F_\alpha$ and $F_\beta$ be two sets of features. We use feature-based abstractions to lift problem-specific characteristics like object names and numbers in order to facilitate the learning of generalized knowledge that can be applied to problems irrespective of differences in such characteristics. Given any SSP $P$, we define *state abstraction* as a function $\alpha : F_\alpha, S \to \overline{S}$ that transforms the concrete state space $S$ of $P$ into a finite abstract state space $\overline{S}$. Similarly, *action abstraction* $\beta : F_\beta, S, A \to \overline{A}$ transforms the action space to a finite abstract action space. Typically, $|\overline{S}| \le |S|$ and $|\overline{A}| \le |A|$. We use $\overline{s} = \alpha(F_\alpha, s)$ and $\overline{a} = \beta(F_\beta, s, a)$ to represent abstractions of a concrete state $s$ and action $a$. In this paper, we utilize feature sets automatically derived using canonical abstraction (Sagiv et al., 2002) to compute such feature-based representations of $s$ and $a$. This is described in Sec. 3.1.

## 3 Our Approach

Our objective is to exploit knowledge from solutions of SSP instances with small object counts to learn Generalized Policy Automata (GPAs) that allow effective pruning of the search space of related SSPs with larger object counts. We accomplish this by using solutions to a small set of training instances that are easily solvable using existing SSP solvers, and using feature-based canonical abstractions to learn a GPA that encodes generalized partial policies and serves as a guide to prune the set of policies under consideration. We provide a brief description of canonical abstraction in Sec. 3.1, define GPAs in Sec. 3.2, and describe our process to learn a GPA in Sec. 3.2.1. We then describe our method (Alg. 1) for solving SSPs and state its theoretical properties in Sec. 3.3.

### 3.1 Canonical Abstraction

Canonical abstractions, commonly used in program analysis, have been shown to be useful in generalized planning (Srivastava et al., 2011; Karia and Srivastava, 2021). Canonical abstractions

**Concrete State $s_1$**

$rock(r_x) : r_1, r_2$
$interesting(r_x) : r_2$
$location(l_x) : l_{base}, l_1, l_2$
$base(l_x) : l_{base}$

$rock\text{-}at(r_x, l_x)$

|       | $l_{base}$ | $l_1$ | $l_2$ |
|-------|------------|-------|-------|
| $r_1$ | 0          | 1     | 0     |
| $r_2$ | 1          | 0     | 0     |

**Concrete State $s_2$**

$rock(r_x) : r_1, r_2, r_3, r_4, r_5, r_6$
$interesting(r_x) : r_4, r_5, r_6$
$location(l_x) : l_{base}, l_1, l_2, l_3$
$base(l_x) : l_{base}$

$rock\text{-}at(r_x, l_x)$

|       | $l_{base}$ | $l_1$ | $l_2$ | $l_3$ |
|-------|------------|-------|-------|-------|
| $r_1$ | 0          | 1     | 0     | 0     |
| $r_2$ | 0          | 0     | 1     | 0     |
| $r_3$ | 0          | 0     | 0     | 1     |
| $r_4$ | 1          | 0     | 0     | 0     |
| $r_5$ | 1          | 0     | 0     | 0     |
| $r_6$ | 1          | 0     | 0     | 0     |

$\xrightarrow{\text{Canonical Abstraction}}$

**Abstract State $\overline{s}$**

Roles
$\psi_1 = \{rock\}$
$\psi_2 = \{rock, interesting\}$
$\psi_3 = \{location\}$
$\psi_4 = \{location, base\}$

$rock\text{-}at(\psi_i, \psi_j)$

|          | $\psi_1$ | $\psi_2$ | $\psi_3$ | $\psi_4$ |
|----------|----------|----------|----------|----------|
| $\psi_1$ | 0        | 0        | 0.5      | 0        |
| $\psi_2$ | 0        | 0        | 0        | 1        |
| $\psi_3$ | 0        | 0        | 0        | 0        |
| $\psi_4$ | 0        | 0        | 0        | 0        |

Figure 1: An example of how canonical abstraction can be used to lift problem-specific characteristics like object names and numbers. $s_1$ and $s_2$ are example states of two *different* problems.

group together multiple objects in a state using object roles. Given a concrete state $s$ and an object $o$, the set of unary predicates that $o$ satisfies is known as the *role* of $o$. 0-ary predicates are represented as unary predicates with a default 'phantom' object.

Let $\psi$ be a role. Then we define $\varphi_\psi(s)$ as a function that returns the set of objects that map to $\psi$ in a concrete state $s$. Similarly, for any given predicate $p_n \in \mathcal{P}$ where $n$ is the arity, $\varphi_{p_n(\psi_1, \dots, \psi_n)}(s)$ is defined as the set of all $n$-ary predicates in $s$ that are consistent with the roles composing the predicate $p_n(\psi_1, \dots, \psi_n)$, i.e., $\varphi_{p_n(\psi_1, \dots, \psi_n)}(s) = \{p_n(o_1, \dots, o_n) | p_n(o_1, \dots, o_n) \in s, o_i \in \varphi_{\psi_i}(s)\}$.

The value of a role $\psi$ in a concrete state $s$ is given as $\max(2, |\varphi_\psi(s)|)$ to indicate whether there are 0, 1, or more than 1 objects satisfying $\psi$. Since relations between objects become imprecise when grouped as roles, the value of $p_n(\psi_1, \dots, \psi_n)$ in $s$ is determined using three-valued logic and is 0 if $\varphi_{p_n(\psi_1, \dots, \psi_n)}(s) = \{\}$, 1 if $|\varphi_{p_n(\psi_1, \dots, \psi_n)}(s)| = |\varphi_{\psi_1}(s)| \times \dots \times |\varphi_{\psi_n}(s)|$, and $\frac{1}{2}$ otherwise.

Let $\Psi$ be the set of all possible roles and $\mathcal{P}_i$ be the set of predicate names $p \in \mathcal{P}$ of arity $i$ for a domain $\mathcal{D}$, then $\overline{\mathcal{P}}_i = \mathcal{P}_i \times [\Psi]^i$ is the set of all possible relations of arity $i$ between roles. We define the feature set for state abstraction as $F_\alpha = \Psi \cup_{i=2}^N \overline{\mathcal{P}}_i$ where $N$ is the maximum arity of any predicate in $\mathcal{D}$. We define state abstraction $\alpha(F_\alpha, s)$ for a given concrete state $s$ to return an abstract state $\overline{s}$ as a total valuation of $F_\alpha$ using the process described above. Similarly, we define the feature set for action abstraction as $F_\beta = \Psi$. The action abstraction $\beta(F_\beta, s, a)$ for a concrete action $a(o_1, \dots, o_n)$ when applied to $s$ returns an abstract action $\overline{a}(\psi_1, \dots, \psi_n)$ where the action name $\overline{a} \equiv a$ and $\psi_i$ is the role of the parameter $o_i$, i.e., $o_i \in \varphi_{\psi_i}(s)$ for any $\psi_i \in \Psi$.

Fig. 1 provides an intuitive example of how canonical abstraction can be used to lift instance-specific information. The figure describes two concrete states $s_1$ and $s_2$ from two different problems. $s_1$ contains 2 rocks and 3 locations whereas $s_2$ contains 6 rocks and 4 locations. There are four roles in $\overline{s}$. For example, in $s_1$, $r_1$ maps to the the role for a *rock* $\psi_1$, and $r_2$ maps to the role for an *interesting rock* $\psi_2$, i.e., $\varphi_{\psi_1}(s_1) = \{r_1\}$ and $\varphi_{\psi_2}(s_1) = \{r_2\}$. The abstract relation *rock-at*$(\psi_i, \psi_j)$ provides the three-valued representation of values between different roles. For example, *rock-at*$(\psi_1, \psi_3)$ is interpreted as *the set of rocks that are at some location*, while *rock-at*$(\psi_2, \psi_4)$ is interpreted as *the set of interesting rocks that are at the base*. Since *rock-at*$(r_1, l_2)$ does not appear in $s_1$, *rock-at*$(\psi_1, \psi_3)$ evaluates to 0.5 to indicate that there are some *rocks* that are at some, but not all, *locations*. Similarly, since all *interesting rocks* are at the base location, *rock-at*$(\psi_2, \psi_4)$ evaluates to 1.

The key aspect of abstraction comes from the observation that the relation *rock-at*$(\psi_i, \psi_j)$ remains the same for $s_2$ even though $s_2$ has many more objects than $s_1$. The same high-level interpretations of the relations are captured while lifting low-level information like object names and numbers.

## 3.2 Generalized Policy Automata

We introduce Generalized Policy Automata (GPAs) as compact and expressive non-deterministic finite-state automata that encode generalized knowledge and can be represented as directed hypergraphs.

GPAs impose hierarchical constraints on the state space of an SSP and prune the action space under consideration, thus reducing the computational effort of solving larger related SSP instances.

**Definition 3.1** (Generalized Policy Automaton). Let $\overline{S}$ and $\overline{A}$ be a set of *abstract* states and actions. A Generalized Policy Automaton (GPA) $\overline{\mathcal{G}} = \langle \overline{\mathcal{V}}, \overline{\mathcal{E}} \rangle$ is a non-deterministic finite-state automaton that can be represented as a directed hypergraph where the set of vertices $\overline{\mathcal{V}} = \overline{S}$. $\overline{\mathcal{E}} \subseteq \overline{\mathcal{V}} \times \mathbb{P}(\overline{\mathcal{V}}) \setminus \varnothing \times \overline{A}$, where $\mathbb{P}(\overline{\mathcal{V}})$ is the powerset of $\overline{\mathcal{V}}$, is a set of directed hyperedges s.t. each hyperedge $\overline{e} \in \overline{\mathcal{E}}$ is a tuple $(\overline{e}_{src}, \overline{e}_{dest}, \overline{e}_{act})$ representing a start vertex, a set of result vertices, and an action label.

**Definition 3.2** (GPA Consistent Policy). A policy $\pi$ for an SSP $P$ is defined to be consistent with a GPA $\overline{\mathcal{G}}$ iff for any states $s, s' \in S$ and any action $a \in A$ whenever $\pi(s) = a$ and $t(s, a, s') > 0$ there exists a hyperedge $\overline{e} \equiv (\overline{e}_{src}, \overline{e}_{dest}, \overline{e}_{act}) \in \overline{\mathcal{E}}$ where $\overline{s} = \overline{e}_{src}$, $\overline{s'} \in \overline{e}_{dest}$, and $\overline{a} = \overline{e}_{act}$.

**Definition 3.3** (Hierarchically Optimal Policy). A policy $\pi$ given a GPA $\overline{\mathcal{G}}$ is hierarchically optimal for an SSP $P$ iff $\pi$ is minimal among all possible policies $\pi'$ that are GPA consistent w.r.t. $\overline{\mathcal{G}}$, i.e., $V^{\pi}(s_0) \leq V^{\pi'}(s_0)$.

### 3.2.1 Learning GPAs

It is well-known that solutions to small problems can be used to construct generalized control structures that can assist in solving larger problems. We adopt a similar strategy of the learn-from-small-examples approach (Wu and Givan, 2007; Karia and Srivastava, 2021, 2022) and compute GPAs iteratively from a small training set containing solutions of similar SSP instances.

To form our training set, we use a library of solution policies $\Pi = \{\pi_1, \ldots, \pi_n\}$ for *small* SSPs $P_1, \ldots, P_n$ that can be easily (and optimally) computed by existing SSP solvers. We use the transition function for $P_i$ to convert each pol-

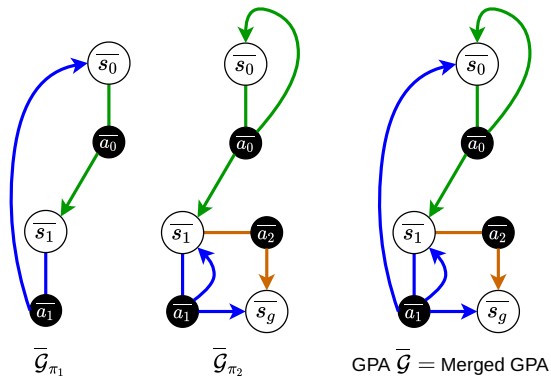

Figure 2: A high-level overview of how we merge different GPAs. All edges with the same color represent a hyperedge. For example, the blue colored hyperedge in $\overline{\mathcal{G}}$ is $(\overline{s_1}, \{\overline{s_0}, \overline{s_1}, \overline{s_g}\}, \overline{a_1})$.

icy $\pi_i \in \Pi$ to a set of transitions $\tau_i = \{(s, a, s') | s, s' \in S_i, a \in A_i, \pi_i(s) = a, t_i(s, a, s') > 0\}$. We then construct our training set $\mathcal{T} = \tau_1 \cup \ldots \cup \tau_n$ as a set of concrete transitions.

The GPA $\overline{\mathcal{G}}_{\Pi}$ learned from a finite set of concrete transitions $\mathcal{T}$ is defined as follows. We first initialize a GPA $\overline{\mathcal{G}}_{\Pi} = \langle \{\}, \{\} \rangle$. Next, we convert $\mathcal{T}$ into an abstract transition set $\overline{\mathcal{T}} = \{(\alpha(F_{\alpha}, s), \beta(F_{\beta}, s, a), \alpha(F_{\alpha}, s')) | (s, a, s') \in \mathcal{T}\}$. We then form the vertex set by using all abstract states in $\overline{\mathcal{T}}$, i.e., $\overline{\mathcal{V}} = \overline{\mathcal{V}} \cup \{\overline{s}, \overline{s'}\}$ for every $(\overline{s}, \overline{a}, \overline{s'}) \in \overline{\mathcal{T}}$. Similarly, we convert each abstract transition into a hyperedge and add it to the edge set, i.e. $\overline{\mathcal{E}} = \overline{\mathcal{E}} \cup (\overline{s}, \{\overline{s'}\}, \overline{a})$ for every $(\overline{s}, \overline{a}, \overline{s'}) \in \overline{\mathcal{T}}$. Finally, we compress $\overline{\mathcal{G}}_{\Pi}$ by replacing edges in $\overline{\mathcal{E}}$ that have the same start nodes and labels but different destinations with a single edge that combines the destinations of the edges, i.e., for any two edges $\overline{e^1}, \overline{e^2} \in \overline{\mathcal{E}}$ s.t. $\overline{e^1}_{src} = \overline{e^2}_{src}$, $\overline{e^1}_{dest} \neq \overline{e^2}_{dest}$, and $\overline{e^1}_{act} = \overline{e^2}_{act}$, $\overline{\mathcal{E}} = \overline{\mathcal{E}} \setminus \{\overline{e^1}, \overline{e^2}\} \cup \{(\overline{e^1}_{src}, \overline{e^1}_{dest} \cup \overline{e^2}_{dest}, \overline{e^1}_{act})\}$. Fig. 2 provides a high-level overview of our procedure for merging GPAs. Henceforth, we drop the subscript $\Pi$ from $\overline{\mathcal{G}}_{\Pi}$ when it is clear from context.

## 3.3 Solving SSPs using GPAs

The key intuition behind our method is to use the GPA to prune policies that are not consistent with the GPA from the search process. We accomplish this by solving a GPA constrained problem that satisfies the constraints encoded by the GPA. We define a GPA constrained problem as follows:

**Definition 3.4** (GPA constrained problem). Let $P = \langle O, S, A, T, C, s_0, g \rangle$ be an SSP for a domain $\mathcal{D}$ and let $\overline{\mathcal{G}} = \langle \overline{\mathcal{V}}, \overline{\mathcal{E}} \rangle$ be a GPA. A GPA constrained problem $P|_{\overline{\mathcal{G}}} = \langle O, S, A, T, C', s_0, g \rangle$ is defined with a cost function $C' : S \times A \times S \to \mathbb{R}^+$ such that $C'[s, a, s'] = C[s, a, s']$ when there exists a

---

**Algorithm 1** GPA acceleration for SSPs

---

**Require:** SSP $P = \langle O, S, A, T, C, s_0, g \rangle$, GPA $\overline{\mathcal{G}} = \langle \overline{\mathcal{V}}, \overline{\mathcal{E}} \rangle$,
  Feature Sets $F_\alpha, F_\beta$, Abstraction Functions $\alpha, \beta$
1: $C' = C$ {copy over the cost function of $P$}
2: **for** $(s, a, s') \in S \times A \times S$ **do**
3:   $\overline{s}, \overline{a}, \overline{s'} \leftarrow \alpha(F_\alpha, s), \beta(F_\beta, s, a), \alpha(F_\alpha, s')$
4:   **if** there is no edge $\overline{e} \in \overline{\mathcal{E}}$ s.t. $\overline{e}_{src} = \overline{s}, \overline{s'} \in \overline{e}_{dest}$, and $\overline{e}_{act} = \overline{a}$ **then**
5:     $C'[s, a, s'] = \infty$
6:   **end if**
7: **end for**
8: $P' = \langle O, S, A, T, C', s_0, g \rangle$
9: $V^*_{P'} \leftarrow$ initializeValueFunction() {Typically using heuristics or randomly}
10: $V^*_{P'}, \pi^*_{P'} \leftarrow$ optimallySolveSSP($P', V^*_{P'}$) {Using the Bellman optimality equation}
11: **if** $\pi^*_{P'}$ is a partial proper policy **then**
12:   **return** $\pi^*_{P'}$ {Return constrained policy}
13: **else**
14:   $V_P, \pi_P \leftarrow$ optimallySolveSSP($P, V^*_{P'}$) {Using the Bellman optimality equation}
15:   **return** $\pi_P$ {Return partial proper policy for $P$}
16: **end if**

---

hyperedge $\overline{e} \equiv (\overline{e}_{src}, \overline{e}_{dest}, \overline{e}_{act}) \in \overline{\mathcal{E}}$ where $\overline{s} = \overline{e}_{src}$, $\overline{s'} \in \overline{e}_{dest}$, and $\overline{a} = \overline{e}_{act}$ and $C'[s, a, s'] = \infty$ if there is no such hyperedge.

The goal of modifying the cost function is to prevent concrete transitions whose abstract translations are absent in the GPA from being used when performing Bellman updates for $P|_{\overline{\mathcal{G}}}$. Actions belonging to such transitions cannot appear in $\pi^*_{P|_{\overline{\mathcal{G}}}}$ since their costs would be $\infty$. As a result, $P|_{\overline{\mathcal{G}}}$ is not an SSP and the existence of a partial proper policy is not guaranteed in $P|_{\overline{\mathcal{G}}}$. Note that every optimal policy $\pi^*_{P|_{\overline{\mathcal{G}}}}$ for $P|_{\overline{\mathcal{G}}}$ is *hierarchically optimal* for an SSP $P$ given a GPA $\overline{\mathcal{G}}$. Our overall objective is to compute such hierarchically optimal policies in a way such that they are approximately as cost effective as the optimal policy while requiring a fraction of the computational effort.

Given a GPA $\overline{\mathcal{G}}$, Alg. 1 works as follows. Lines 1-8 create a GPA constrained problem $P'$. Next, line 10 optimally solves $P'$ using any off-the-shelf SSP solver with a randomly initialized value function (line 9). If the computed policy $\pi^*_{P'}$ is a partial proper policy then it is returned immediately (lines 11-12) else Alg. 1 uses a new instance of the SSP solver to compute a policy for $P$ using $V^*_{P'}$ as the *bootstrapped* initial value function. Information such as whether a state is a dead end, etc., is not carried over. Line 14 then optimally solves $P$ using $V^*_{P'}$ as the initial value estimates and returns a partial proper policy $\pi_P$ for $P$. Since $V^*_{P'}$ is only used as an initial bootstrapping estimate for $P$, an SSP solver will only return a policy $\pi_P$ that is better than or equal to $\pi^*_{P'}$ following standard results on policy improvement for value iteration (Sutton and Barto, 1998). The following result shows that Alg. 1 computes hierarchically optimal policies.[1]

**Theorem 1.** When Alg. 1 returns a constrained policy $\pi^*_{P'}$ (line 12) for an SSP $P$ given a GPA $\overline{\mathcal{G}}$ then $\pi^*_{P'}$ is hierarchically optimal.

*Proof* (Sketch). Intuitively, Alg. 1 creates a new problem $P'$ (lines 1-8) using the original cost function of $P$. The cost function for a concrete transition $(s, a, s') \in S \times A \times S$ is set to $\infty$ iff the corresponding abstract transition is not part of a hyperedge in $\overline{\mathcal{G}}$. As a result, $P' \equiv P|_{\overline{\mathcal{G}}}$, and any partial policy $\pi_{P'}$ for $\overline{\mathcal{G}}$ is a GPA consistent policy. Since Alg. 1 optimally solves $P'$ (line 10), the returned policy is hierarchically optimal. $\square$

Alg. 1 computes $\pi^*_{P|_{\overline{\mathcal{G}}}}$ for $P|_{\overline{\mathcal{G}}}$ in the space of cross-product of the states of the GPA $\overline{\mathcal{G}}$ with the states of $P$, similar to HAMs (Parr and Russell, 1997). As seen in our empirical analysis in Sec. 4, we observe that a small set of example policies are sufficient to capture rich generalized control structures that are encoded by such hierarchically optimal policies. Alg. 1 computes such policies within a fraction of the original computational effort and in most cases with costs comparable to $\pi^*_P$.

---

[1]Please see Appendix A.1 for the complete proof.

**Theorem 2.** Given a GPA $\overline{\mathcal{G}}$ and an SSP $P$, Alg. 1 always returns a partial proper policy.

*Proof.* Alg. 1 only returns if it computes a partial proper policy at line 12 or a partial proper policy for $P$ at line 15, which by definition always exists. $\square$

The next result indicates that the output of Alg. 1 is hierarchically optimal or better in terms of the expected value at $s_0$.

**Theorem 3.** Let $V^\pi$ be the value function for a policy $\pi$ returned by Alg. 1 for an SSP $P$ using GPA $\overline{\mathcal{G}}$. Let $V^*_{P|_{\overline{\mathcal{G}}}}$ be the optimal value function for $P|_{\overline{\mathcal{G}}}$, then $V^\pi(s_0) \leq V^*_{P|_{\overline{\mathcal{G}}}}(s_0)$.

*Proof.* If Alg. 1 finds a partial proper policy $\pi^*_{P|_{\overline{\mathcal{G}}}}$ for $P|_{\overline{\mathcal{G}}}$ then it returns it immediately (lines 11-12), in which case $V^\pi(s_0) = V^*_{P|_{\overline{\mathcal{G}}}}(s_0)$. If $\pi^*_{P|_{\overline{\mathcal{G}}}}$ is not a partial proper policy then $V^*_{P|_{\overline{\mathcal{G}}}}(s_0) = \infty$. Since Alg. 1 is complete (Thm. 2), $\pi$ is a partial proper policy where $V^\pi(s_0) < \infty$. $\square$

**Corollary 3.1.** If $V^*_P(s_0) = V^*_{P|_{\overline{\mathcal{G}}}}(s_0)$, then Alg. 1 returns the optimal policy for $P$.

The following result indicates that only a finite set of training examples are needed to learn a GPA such that the constrained problem will always yield the optimal policy for a given domain $\mathcal{D}$.

**Theorem 4.** Suppose $\mathcal{D}$ is a domain, $g$ is a formula over the predicates in $\mathcal{D}$'s vocabulary, $\overline{\mathcal{G}}_{\Pi^*}$ is a GPA s.t. for every SSP instance $P$ of $\mathcal{D}$ whose goal is $g$, there exists an optimal policy $\pi^*_P$ that is consistent with $\overline{\mathcal{G}}_{\Pi^*}$. Then there exists a finite set of policies $\Pi^*$ from which $\overline{\mathcal{G}}_{\Pi^*}$ can be learned.

*Proof.* Since the sizes of the abstract state and action spaces $\overline{S}$ and $\overline{A}$ are finite, the size of $\overline{\mathcal{G}}_{\Pi^*}$ is finite and is bounded by $|\overline{S}| \times |\overline{A}| \times |\overline{S}|$. For every $\overline{e} \in \overline{\mathcal{E}}$ at most $|\overline{e}_{dest}|$ different transitions are needed to learn $\overline{e}$. Since there is a finite number of edges, a finite amount of training data will suffice for learning $\overline{\mathcal{G}}_{\Pi^*}$. $\square$

In the worst case, the cost functions of $P$ and $P|_{\overline{\mathcal{G}}}$ are similar and no savings are obtained. However, in our empirical evaluation, we observed that typically very few and small training problems suffice to learn a compact GPA $\overline{\mathcal{G}}$ that allows efficient computation of solutions for problems that are significantly larger than those used during training. Finding the right set of examples $\Pi^*$ from which $\overline{\mathcal{G}}_{\Pi^*}$ can be learned is an interesting and non-trivial research problem that we leave to future work.

## 4 Experiments

We conducted an empirical evaluation on five well-known benchmark domains that were selected from the International Planning Competition (IPC) (Long and Fox, 2003), International Probabilistic Planning Competition (IPPC) (Younes et al., 2005), and robotic planning (Shah et al., 2020). As a part of our analysis, we use the time required to compute a solution and measure the quality of the solutions found to determine whether GPAs allow efficient solving of SSPs.

We chose PPDDL as our representational language, which was the default language in IPPCs until 2011, after which the Relational Dynamic Influence Diagram Language (RDDL) (Sanner, 2010) became the default. We chose PPDDL over RDDL since modern state-of-the-art solvers for PPDDL are available and since RDDL does not allow specifying the goal condition easily.

For our baselines, we focus on complete solvers for SSPs. Deep Learning based approaches do not guarantee completeness and thus are not directly comparable with our work. We used Labeled RTDP (LRTDP) (Bonet and Geffner, 2003) and Soft-FLARES (Pineda and Zilberstein, 2019) which are state-of-the-art (SOA), complete SSP solvers. These algorithms internally generate their own heuristics for initializing the value function using the input domain and problem file. We used the inadmissible FF heuristic (Hoffmann, 2001) as the internal heuristic function for all algorithms since the baselines performed best using it.

We ran our experiments on a cluster of Intel Xeon E5-2680 v4 CPUs running at 2.4 GHz with 16 GiB of RAM. Our implementation is a Python adaptation of `mdp-lib` (Pineda and Zilberstein, 2019).[2] We utilized problem generators from the IPC and IPPC suites and those in Shah et al. (2020) for

---

[2]Our source code is available at `https://github.com/AAIR-lab/GRAPL`

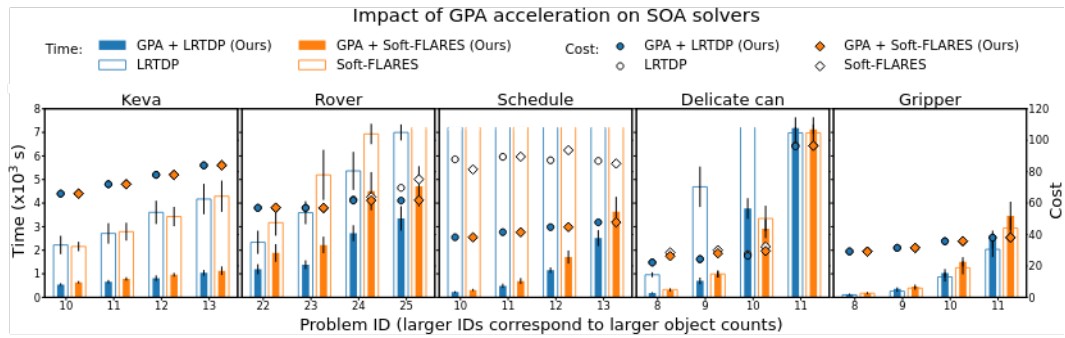

Figure 3: Impact of learned GPAs on solver performance (lower values better). Left y-axes and bars show solution times (in units of 1000 secs) for our approach and baseline SOA solvers (LRTDP and Soft-FLARES). Right y-axes and points show cost incurred by the policy computed by our approach and the baselines. We use the same SSP solver as the corresponding baseline in our approach. Error bars indicate 1 standard deviation (SD) averaged across 10 runs. For clarity, we only report results for the largest test problems and omit error bars from costs due to the low SDs. Open bars at the top indicate a timeout. A complete view of our empirical results is available in Appendix B.

generating the training and test problems for all domains. We provide a brief description of the problem domains below.

**Rover**$(r, w, s, o)$ A set of $r$ rovers need to take images of $o$ objectives, and pickup and drop $s$ samples that are present at one of $w$ waypoints. This is an IPC domain that was converted to a stochastic version by modifying the pickup action to fail with a probability of $0.4$ (leaving the state unchanged).

**Gripper**$(b)$ A robot with two grippers is placed in an environment consisting of two rooms A and B. The objective of the robot is to transfer all the $b$ balls initially located in room A to room B. We modified the gripper to be slippery so that picking a ball has a 20% chance of failure.

**Schedule**$(C, p)$ is an IPPC domain that consists of a set of $p$ packets, each belonging to one of $C$ different classes that need to be queued. A router must first process the arrival of a packet in order to route it. The interval at which the router processes arrivals is determined by probability 0.94.

**Keva**$(P, h)$ A robot uses $P$ keva planks to build a tower of height $h$. Planks are placed in a specific order in one of two locations, preferring one location with probability $0.6$. Despite this simple setting, the Keva domain has been shown to be a challenging problem in robotics (Shah et al., 2020).

**Delicate Can**$(c)$ An arrangement of $c$ cans on a table of which one is a delicate can. The objective is to pick up a specific goal can. Cans can obstruct the trajectory to the goal can and need to be moved. They can be crushed with probability 0.1 (delicate cans with probability 0.8) and need to be revived.

**Training Setup** Our method learns GPAs in a few-shot fashion requiring little to no training data. For forming our training set, we used a minimum of three and a maximum of ten different solution policies (obtained using LAO* (Hansen and Zilberstein, 2001)) for each domain. The time required to learn a GPA was less than 10 seconds in all cases of our experiments. Our training speed highlights the advantages of GPAs that can be quickly learned in a few-shot setting. Moreover, compared to neuro-symbolic methods, GPAs are not subject to catastrophic forgetting and new training data can easily be merged with the existing GPA using the process described in Sec. 3.2.1.

**Test Setup** We fixed the time and memory limit for each problem to 7200 seconds and 16 GiB respectively. To demonstrate generalizability, our test set contains problems with object counts that are much larger than the training policies used. The largest problems in our test sets contain at least twice the number of objects than those used during training. For example, in the Keva domain, we use training policies with towers of height up to 6 and evaluate on problems with towers of height up to 14. The minimum and maximum number of problems that we used in our test set are 12 and 26 problems respectively. Additional information of our empirical setup such as problem parameters, hyperparameters used for configuring baselines, etc., is included in Appendix B.

## 4.1 Results and Analysis

Our evaluation metric compares the time required to find a partial proper policy. We also compare the quality of the computed policies by executing the policies for 100 trials with a horizon limit of

100 and averaging the obtained costs. We report our overall results averaged across 10 different runs and report results up to one standard deviation. Results of our experiments are illustrated in Fig. 3.

In four out of five domains (Schedule, Rover, Keva, and Delicate Can), our approach takes significantly less time compared to the corresponding baseline. For example, in Schedule, the baselines timed out for all of the large test problems reported. GPAs are able to successfully prune away action transitions that are not helping, leading to large savings in the computational effort. The costs obtained for executing these policies are also quite similar to each baseline (e.g., Keva), showing that GPAs are capable of learning good policies much faster without compromising solution quality.

Our approach was unable to outperform the baselines in the Gripper domain. An interesting phenomenon that we observed was that training policies returned by LAO* were different for the case of even/odd balls due to tie-breaking and this led to the GPA not pruning actions as effectively. Nevertheless, we expected the GPA to outperform the baselines. We performed a deeper investigation and found that the FF heuristic used is already well-suited to prune away actions that the GPA would have otherwise pruned. This results in additional overhead being added in our SSP solver from the process of abstraction. However, heuristics that allow such pruning are difficult to synthesize, and in many cases, are hand-coded by an expert after employing significant effort.

There was a decrease in the gains observed in the Delicate Can domain between problem IDs 10 and 11. We investigated this issue and found that the GPA constructed was unable to effectively prune transitions in problem ID 11. This is because the training policies that were used were not sufficient to learn a GPA that could generalize well to larger problems in the Delicate Can domain. As a result, $\pi^*_{P|_{\overline{\mathcal{G}}}}$ was not a partial proper policy for problem ID 11 and thus additional computation was required.

Finally, because of the fixed timeout used, the maximum time of the baselines was bounded, making the impact of GPAs appear smaller in larger problems. For example, in problem ID 25 (10) of the Rover (Delicate Can) domain, the Soft-FLARES (LRTDP) baseline timed out in all our runs, but it had found a policy that had comparable costs. However, when allowed to run to convergence, it took over 15000 seconds in a targeted experiment that we performed for investigating this issue.

## 5   Related Work

There has been plenty of dedicated research to improve the efficiency for solving SSPs. LAO* (Hansen and Zilberstein, 2001) computes policies by using heuristics to guide the search process. LRTDP (Bonet and Geffner, 2003) uses a labeling procedure in RTDP wherein a part of the subtree that is $\epsilon$-consistent is marked as *solved* leading to faster ending of trials. SSiPP (Trevizan and Veloso, 2012) uses short-sightedness by only considering reachable states up to $t$ states away and solving this constrained SSP. Soft-FLARES (Pineda and Zilberstein, 2019) combines labeling and short-sightedness for computing solutions. These approaches are complete and can be configured to return optimal solutions, however, they do not learn any generalized knowledge and as result cannot readily scale to problems with a larger number of objects.

Determinization-based approaches (Yoon et al., 2007; Pineda and Zilberstein, 2014) build sparse representations of SSP problems by reducing the branching factor in the "environment's choices" (the set of probabilistic effects of an action), while our approach uses abstraction to create abstract controllers that generalize solutions to SSPs and reduces the branching factor in the agent's choice (the set of applicable actions). Our approach always considers all possible outcomes of every action. This is a key advantage of our approach since GPAs are able to better handle unexpected outcomes when executing an action in the policy that would otherwise require replanning.

Boutilier et al. (2001) utilize decision-theoretic regression to compute generalized policies for first-order MDPs represented using situation calculus. They utilize symbolic dynamic programming to compute a symbolic value function that applies to problems with varying number of objects. FOALP (Sanner and Boutilier, 2005) uses linear programming to compute an approximation of the value function for first-order MDPs while providing upper bounds on the approximation error irrespective of the domain size. A key limitation of their approach is requiring the use of a representation of action models over which it is possible to regress using situation calculus. API (Fern et al., 2006) uses approximate policy iteration with taxonomic decision lists to form policies. They use Monte Carlo simulations with random walks on a single problem to construct a policy. API offers no guarantees of completeness or hierarchical optimality.

Parr and Russell (1997) propose the hierarchical abstract machine (HAM) framework wherein component solutions from problem instances can be combined to solve larger problem instances efficiently. Recently, Bai and Russell (2017) extended HAMs to Reinforcement Learning (RL) settings − where transition dynamics are not known − by leveraging internal transitions of the HAMs. A key limitation of both these approaches is that the HAMs were hand-coded by a domain expert.

Related work in Generalized Planning focuses on the problem of computing generalized plans and policies such as our GPAs (Srivastava et al., 2012). Bonet et al. (2009) automatically create finite-state controllers for solving problems using a set of examples by modeling the search as a contingent problem. Their approach is limited in applicability since it only works on deterministic problems and the features they use are hand-coded. Aguas et al. (2016) utilize small example policies to synthesize hierarchical finite state controllers that can call each other. However, their approach requires all training data to be provided upfront. D2L (Francès et al., 2021) utilizes description logics to automatically generate features and reactive policies based on those features. Their approach comes with no guarantees for finding a solution and can only work on deterministic problems.

Deep Learning based approaches such as ASNets (Toyer et al., 2018) learn generalized policies for SSPs using a neuro-symbolic approach. They use the action schema from PPDDL to create alternating action and proposition layers. They do not learn generalized controllers and instead duplicate weights in a post-processing step to represent the generalized policy. GRL (Karia and Srivastava, 2022) uses Description Logic based feature evaluations of states as input to a neural network for computing generalized policies in RL settings. A common limitation of such approaches is the lack of any interpretability as well as theoretical guarantees of completeness or optimality. Furthermore, these approaches can be susceptible to catastrophic forgetting. GPAs are interpretable, provide strong theoretical guarantees, and do not lose any information when training data is presented as a stream.

Our approach differs from existing work in several aspects. Our approach constructs a domain-dependent GPA automatically without any human intervention. Using canonical abstraction, we lift problem-specific characteristics like object names and object counts in a domain-independent fashion. Another key difference between other techniques is that our approach can easily incorporate solutions from new examples into the GPA without having to remember any of the earlier examples. This allows our learning to scale better and can naturally utilize *leapfrog learning* (Groshev et al., 2018; Karia and Srivastava, 2021, 2022) when presented with a large problem in the absence of training data. Finally, our approach comes with guarantees of completeness and hierarchical optimality given the training data presented. This means that if a solution exists, our approach will find it (Thm. 2) and it will be guaranteed to be hierarchically optimal or better (Thm. 3).

# 6   Conclusions and Future Work

We show that non-deterministic Generalized Policy Automata (GPAs) constructed using solutions of small example SSPs are able to significantly reduce the computational effort for finding solutions for larger related SSPs. Furthermore, for many benchmark problems, the search space pruned by GPAs does not prune away relevant transitions thus allowing our approach to compute policies of comparable cost in a fraction of the effort. Our approach comes with guarantees of hierarchical optimality and also comes with the guarantee of always finding a solution to the SSP.

There are several interesting research directions for future work. Description Logics are more expressive than canonical abstractions and have been demonstrated to be effective at synthesizing memoryless controllers for deterministic planning problems (Bonet et al., 2019). Our approach can easily utilize any relational abstraction and it would be interesting to evaluate the efficacy of description logics. Finally, our approach is applicable when solutions have a pattern. We believe that more intelligent training data generation methods could help improve performance in domains like Delicate Can. We plan to investigate these directions of research in future work.

## Acknowledgements

We would like to thank Deepak Kala Vasudevan for help with a prototype implementation of the source code. We would like to thank the Research Computing Group at Arizona State University for providing compute hours for our experiments. This work was supported in part by the NSF under grants IIS 1909370 and IIS 1942856.

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
