# A Proofs

**Definition A.1** (Transition Graph of a Policy). Given an SSP $P$ and a policy $\pi$, the transition graph $G_\pi$ for $\pi$ is defined as a tuple $\langle V, E, L \rangle$ where $V \equiv S$ and for every transition $(s, a, s') \in S \times A \times S$, $(s, s') \in E$ and $L(s, s') = a$ if and only if $\pi(s) = a$.

Given a policy $\pi$, the transition graph $G_\pi$ for an SSP $P$ can be easily constructed by starting at $s_0$ and using the transition function to add all nodes and edges that are reachable from $s_0$ using $\pi$. The cost of an edge $(s, s')$ is $c(s, L(s, s'), s')$ and it can be easily extracted using the cost function of $P$.

## A.1 Proof for Theorem 1

**Theorem 1.** When Alg. 1 returns a constrained policy $\pi_{P'}^*$ (line 12) for an SSP $P$ given a GPA $\overline{\mathcal{G}}$ then $\pi_{P'}^*$ is hierarchically optimal.

*Proof.* We will prove that when Alg. 1 uses $\overline{\mathcal{G}}$ to set costs of concrete transitions to infinity (lines 1-8) it effectively prunes out any transitions that are not consistent with $\overline{\mathcal{G}}$. We prove this by proving that, (a) any policy returned by Alg. 1 cannot use an infinity transition, and (b) every policy that is consistent with $\overline{\mathcal{G}}$ will be considered by Alg. 1.

We first show that any policy returned by Alg. 1 cannot use an transition whose cost is infinity. When Alg. 1 returns a policy $\pi_{P'}^*$ at line 12 it is a partial proper policy since line 11 checks if $\pi_{P'}^*$ is a partial proper policy. As a result, we can argue that $\pi_{P'}^*$ cannot have any infinity edges in the transition graph. Suppose that there was an infinity edge $(s, s')$ in the transition graph of $\pi_{P'}^*$. Since the action $a \equiv L(s, s')$ has been included in the policy, the value of the state $s$ where $\pi(s) = a$ is infinity, i.e., $V_{P'}^*(s) = \infty$. Since Alg. 1 uses the Bellman optimality equation: $V^*(s_0) = \min_{a \in A} \sum_{s' \in S} t(s_0, a, s')[c(s_0, a, s') + V^*(s')]$ (Sec. 2) for computing a policy for $P'$ (line 10) and since $s$ is reachable from $s_0$, $V(s_0)$ would also be $\infty$. However, this is a contradiction since $\pi_{P'}^*$ is a partial proper policy and therefore $V_{P'}^*(s_0) < \infty$. This shows that policies considered by Alg. 1 are GPA consistent policies w.r.t. $\overline{\mathcal{G}}$.

We now show that every policy that is consistent with $\overline{\mathcal{G}}$ is considered by Alg. 1. Suppose that there was an optimal policy $\pi_{P'}' \neq \pi_{P'}^*$ that was also consistent with $\overline{\mathcal{G}}$ and was better than $\pi_{P'}^*$, i.e., $V_{P'}^{\pi'}(s_0) < V_{P'}^*(s_0)$. Now, consider every action choice made by $\pi'$ in the transition graph $G_{\pi'}$ rooted at $s_0$. All of these edges are consistent with $\overline{\mathcal{G}}$ and thus are not set as infinite cost by Alg. 1. Thus, the expected cost $V_{P'}^{\pi'}(s_0)$ will be the true expected cost. This is minimal since Alg. 1 uses the Bellman optimality equation that is guaranteed to converge to the minimal value. This means that it must converge to a minimal value that is at most the expected value of $V_{P'}^{\pi'}$ and thus return $\pi_{P'}'$ which is a contradiction. This shows that Alg. 1 cannot miss a "good" policy that is GPA consistent w.r.t. $\overline{\mathcal{G}}$.

Thus, the policy $\pi_{P'}^*$ returned by Alg. 1 is hierarchically optimal w.r.t. all policies $\pi$ that are consistent with the GPA $\overline{\mathcal{G}}$.

$\square$

# B Extended Experiments and Results

**Training and Test Setup** Descriptions of the training problems used by us and their parameters can be found in Table 1. Test problems and parameters along with complete information for the solution times, costs, and their standard deviations for 10 runs are available in tabular format in Tables 2, 3, 4, 5, and 6. Note that for the Keva domain, the standard deviations for the costs incurred are 0. This is accurate since the only source of stochasticity in Keva is a *human place* action that determines where the human places a plank which is one of two locations. As a result, Keva policies are deterministic in execution since the human always places a plank and all other actions are deterministic. It is interesting that despite this simplistic domain, the baselines are unable to reasonably converge within the timeout. We also present an extended version of Fig. 3 of the main paper that includes results for a larger suite a test problems for a better view of our overall results. These results are reported in Fig. 4. Note that the solution times on the left y-axis of these plots are shown in log scale. For better visualization, we omitted the problems with smaller IDs, mainly whose solutions times (in log scale) were not visible for both our as well as baseline approaches.

**Hyperparameters** We used $\epsilon = 10^{-5}$ as the value for determining whether an algorithm has converged to an $\epsilon$-consistent policy. We set the total number of trials/iterations for all algorithms to $\infty$. As a result, each algorithm would only return once it has converged or if the time limit has been exceeded. For Soft-FLARES, we used $t = 4$ which controls the horizon of the sub-tree that is checked for being $\epsilon$-consistent during the labeling procedure. Our distance metric is the *step* function which simply counts the depth until the horizon is exceeded. For the selective sampling procedure, we used the *logistic* sampler configured with $\alpha = 0.1$ and $\beta = 0.9$.

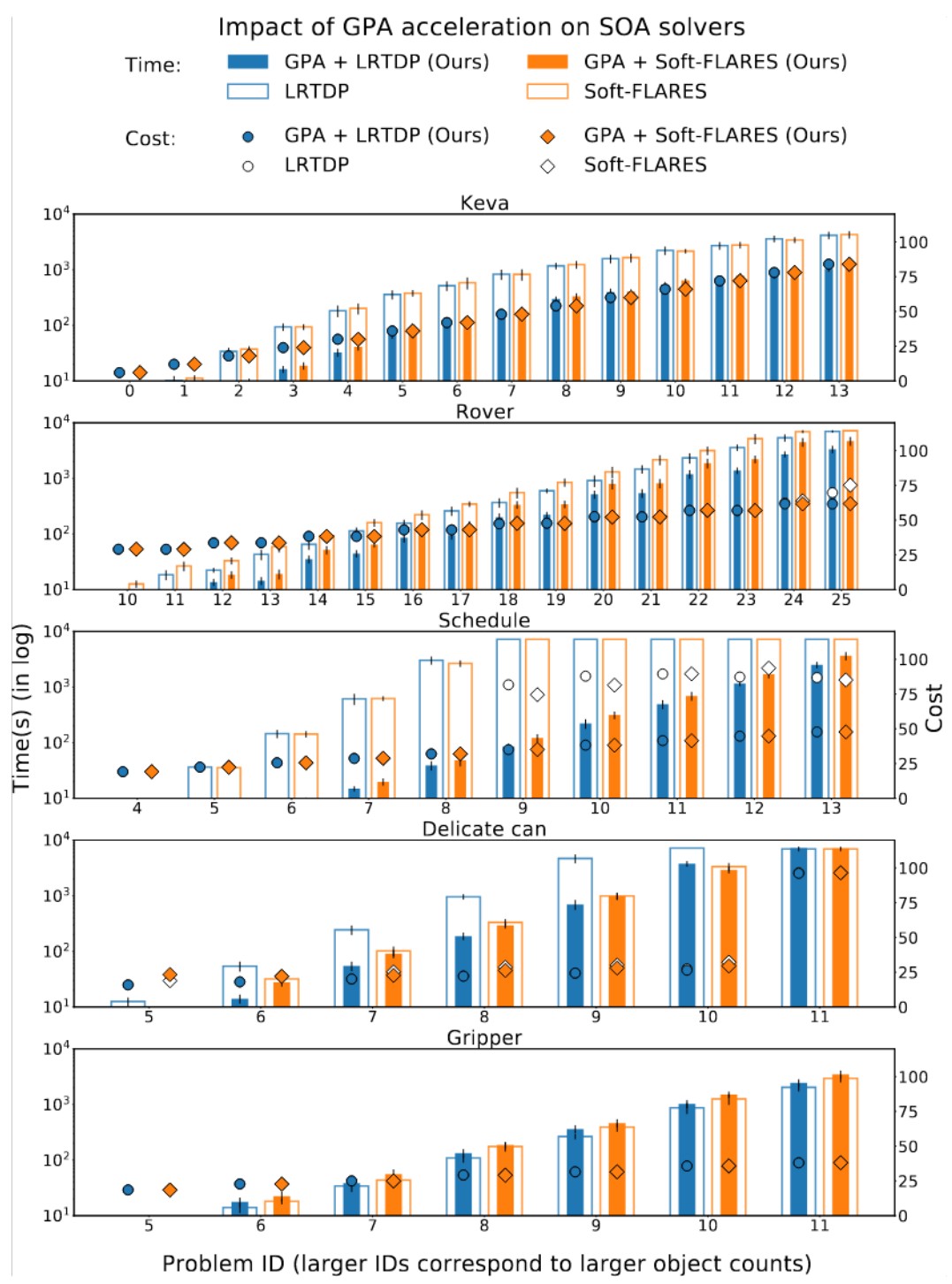

Figure 4: Impact of learned GPAs on solver performance of test problems (lower values better). Left y-axes and bars show solution times (in log scale), and right y-axes and points show cost incurred by the policy computed by our approach and baseline SOA solvers (LRTDP and Soft-FLARES). We use the same SSP solver as the corresponding baseline in our approach. Error bars for solution times indicate 1 standard deviation (SD) averaged across 10 runs.

Table 1: Our training setup for all domains. ID refers to the problem ID in the training set. The other columns refer to the parameters passed to the problem generator for generating the problem. Entries marked − indicate that there was no such problem ID in the training set.

| | | | $\theta$ | | |
|---|---|---|---|---|---|
| ID | Keva$(P, h)$ | Rover$(r, w, s, o)$ | Schedule$(C, p)$ | Delicate Can$(c)$ | Gripper$(b)$ |
| 0 | (2, 1) | (1, 3, 1, 2) | (1, 2) | (2) | (1) |
| 1 | (4, 2) | (1, 4, 1, 2) | (1, 3) | (3) | (1) |
| 2 | (6, 3) | (1, 3, 2, 2) | (1, 4) | (4) | (2) |
| 3 | (8, 4) | (1, 4, 2, 2) | − | (5) | (2) |
| 4 | (10, 5) | (1, 3, 3, 2) | − | (6) | (3) |
| 5 | (12, 6) | (1, 4, 3, 2) | − | − | (3) |
| 6 | − | (1, 3, 4, 2) | − | − | (4) |
| 7 | − | (1, 4, 4, 2) | − | − | (4) |
| 8 | − | (1, 3, 5, 2) | − | − | (5) |
| 9 | − | (1, 4, 5, 2) | − | − | (5) |

Table 2: Our test setup for the Keva$(P, h)$ domain (lower values better). ID refers to the problem ID in the test set. $\theta$ refers to the parameters passed to the problem generator for generating the problem. Times indicate the seconds required to find a policy. Similarly, costs are reported as average costs obtained by executing the computed policy for 100 trials. We ran our experiments using a different random seed for 10 different runs and report average metrics up to one standard deviation. Better metrics are at least 5% better and are indicated using bold font.

| | | Time($x \equiv$ LRTDP) | | Time($x \equiv$ Soft-FLARES) | | Cost($x \equiv$ LRTDP) | | Cost($x \equiv$ Soft-FLARES) | |
|---|---|---|---|---|---|---|---|---|---|
| ID | $\theta$ | Ours + $x$ | $x$ | $x$ | Ours + $x$ | $x$ | Ours + $x$ | $x$ | Ours + $x$ |
| 0 | (29, 1) | **1.44** ±0.17 | 2.45 ±0.32 | 2.65 ±0.37 | **1.47** ±0.27 | 6.00 ±0.00 | 6.00 ±0.00 | 6.00 ±0.00 | 6.00 ±0.00 |
| 1 | (29, 2) | **4.15** ±0.68 | 10.23 ±2.05 | 11.22 ±1.45 | **4.63** ±0.58 | 12.00 ±0.00 | 12.00 ±0.00 | 12.00 ±0.00 | 12.00 ±0.00 |
| 2 | (29, 3) | **8.14** ±0.92 | 34.05 ±5.53 | 37.33 ±4.71 | **9.55** ±1.45 | 18.00 ±0.00 | 18.00 ±0.00 | 18.00 ±0.00 | 18.00 ±0.00 |
| 3 | (29, 4) | **16.26** ±2.34 | 93.78 ±14.53 | 93.39 ±11.30 | **18.80** ±2.84 | 24.00 ±0.00 | 24.00 ±0.00 | 24.00 ±0.00 | 24.00 ±0.00 |
| 4 | (29, 5) | **32.61** ±5.40 | 183.57 ±43.80 | 201.99 ±45.05 | **41.19** ±6.05 | 30.00 ±0.00 | 30.00 ±0.00 | 30.00 ±0.00 | 30.00 ±0.00 |
| 5 | (29, 6) | **68.73** ±11.04 | 358.98 ±67.32 | 379.84 ±51.43 | **74.47** ±9.10 | 36.00 ±0.00 | 36.00 ±0.00 | 36.00 ±0.00 | 36.00 ±0.00 |
| 6 | (29, 7) | **115.28** ±11.90 | 515.85 ±101.56 | 583.52 ±143.15 | **120.30** ±17.58 | 42.00 ±0.00 | 42.00 ±0.00 | 42.00 ±0.00 | 42.00 ±0.00 |
| 7 | (29, 8) | **174.88** ±17.36 | 829.19 ±174.46 | 825.84 ±192.54 | **191.44** ±27.16 | 48.00 ±0.00 | 48.00 ±0.00 | 48.00 ±0.00 | 48.00 ±0.00 |
| 8 | (29, 9) | **298.61** ±30.82 | 1174.79 ±160.45 | 1236.91 ±200.02 | **331.95** ±45.28 | 54.00 ±0.00 | 54.00 ±0.00 | 54.00 ±0.00 | 54.00 ±0.00 |
| 9 | (29, 10) | **394.71** ±62.16 | 1578.38 ±279.31 | 1647.76 ±297.42 | **406.45** ±37.23 | 60.00 ±0.00 | 60.00 ±0.00 | 60.00 ±0.00 | 60.00 ±0.00 |
| 10 | (29, 11) | **544.26** ±58.84 | 2223.43 ±390.05 | 2158.91 ±199.54 | **639.60** ±54.35 | 66.00 ±0.00 | 66.00 ±0.00 | 66.00 ±0.00 | 66.00 ±0.00 |
| 11 | (29, 12) | **665.29** ±76.02 | 2713.78 ±435.39 | 2787.92 ±378.85 | **782.82** ±74.38 | 72.00 ±0.00 | 72.00 ±0.00 | 72.00 ±0.00 | 72.00 ±0.00 |
| 12 | (29, 13) | **815.11** ±117.62 | 3606.12 ±494.69 | 3427.93 ±409.34 | **958.17** ±81.59 | 78.00 ±0.00 | 78.00 ±0.00 | 78.00 ±0.00 | 78.00 ±0.00 |
| 13 | (29, 14) | **1042.44** ±119.63 | 4171.27 ±645.70 | 4291.64 ±663.83 | **1128.84** ±183.24 | 84.00 ±0.00 | 84.00 ±0.00 | 84.00 ±0.00 | 84.00 ±0.00 |

Table 3: Our test setup for the Rover($r, w, s, o$) domain (lower values better). ID refers to the problem ID in the test set. $\theta$ refers to the parameters passed to the problem generator for generating the problem. Times indicate the seconds required to find a policy. Similarly, costs are reported as average costs obtained by executing the computed policy for 100 trials. We ran our experiments using a different random seed for 10 different runs and report average metrics up to one standard deviation. Better metrics are at least 5% better and are indicated using bold font.

| ID | $\theta$ | Time($x \equiv$ LRTDP) | | Time($x \equiv$ Soft-FLARES) | | Cost($x \equiv$ LRTDP) | | Cost($x \equiv$ Soft-FLARES) | |
|---|---|---|---|---|---|---|---|---|---|
| | | $x$ | Ours + $x$ | $x$ | Ours + $x$ | $x$ | Ours + $x$ | $x$ | Ours + $x$ |
| 0 | (1, 3, 1, 2) | 0.02 ±0.01 | 0.02 ±0.01 | 0.02 ±0.01 | **0.01** ±0.00 | 6.62 ±0.08 | 6.68 ±0.08 | 6.69 ±0.12 | 6.68 ±0.09 |
| 1 | (1, 4, 1, 2) | 0.02 ±0.00 | 0.02 ±0.00 | 0.02 ±0.01 | **0.01** ±0.00 | 6.64 ±0.08 | 6.70 ±0.09 | 6.62 ±0.12 | 6.70 ±0.12 |
| 2 | (1, 3, 2, 2) | 0.10 ±0.03 | **0.03** ±0.01 | 0.09 ±0.02 | **0.03** ±0.00 | 10.33 ±0.14 | 10.29 ±0.08 | 10.38 ±0.15 | 10.34 ±0.10 |
| 3 | (1, 4, 2, 2) | 0.18 ±0.04 | **0.03** ±0.01 | 0.22 ±0.02 | **0.04** ±0.01 | 10.34 ±0.11 | 10.30 ±0.12 | 10.27 ±0.12 | 10.26 ±0.16 |
| 4 | (1, 3, 3, 2) | 0.43 ±0.06 | **0.18** ±0.03 | 0.46 ±0.07 | **0.20** ±0.04 | 15.03 ±0.19 | 15.01 ±0.16 | 14.94 ±0.23 | 15.09 ±0.11 |
| 5 | (1, 4, 3, 2) | 0.80 ±0.14 | **0.10** ±0.02 | 0.96 ±0.24 | **0.09** ±0.01 | 15.00 ±0.23 | 15.01 ±0.23 | 14.95 ±0.17 | 14.95 ±0.19 |
| 6 | (1, 3, 4, 2) | 1.08 ±0.12 | **0.54** ±0.10 | 1.66 ±0.39 | **0.68** ±0.11 | 19.76 ±0.21 | 19.65 ±0.31 | 19.68 ±0.15 | 19.62 ±0.17 |
| 7 | (1, 4, 4, 2) | 2.33 ±0.43 | **0.54** ±0.08 | 3.22 ±0.69 | **0.70** ±0.14 | 19.63 ±0.13 | 19.63 ±0.18 | 19.70 ±0.16 | 19.73 ±0.20 |
| 8 | (1, 3, 5, 2) | 3.54 ±0.59 | **1.61** ±0.23 | 4.25 ±0.84 | **2.42** ±0.51 | 24.36 ±0.17 | 24.35 ±0.30 | 24.34 ±0.22 | 24.46 ±0.09 |
| 9 | (1, 4, 5, 2) | 7.78 ±1.62 | **1.82** ±0.28 | 9.57 ±1.69 | **2.26** ±0.42 | 24.41 ±0.27 | 24.38 ±0.16 | 24.32 ±0.34 | 24.23 ±0.32 |
| 10 | (1, 3, 6, 2) | 8.77 ±1.08 | **4.91** ±1.04 | 12.67 ±1.91 | **6.40** ±0.93 | 28.95 ±0.26 | 29.13 ±0.26 | 28.98 ±0.25 | 29.14 ±0.31 |
| 11 | (1, 4, 6, 2) | 18.50 ±3.76 | **4.31** ±0.77 | 26.62 ±5.28 | **5.28** ±0.56 | 28.96 ±0.29 | 29.07 ±0.18 | 29.09 ±0.18 | 28.86 ±0.25 |
| 12 | (1, 3, 7, 2) | 22.42 ±2.26 | **13.60** ±1.98 | 33.11 ±4.79 | **18.49** ±2.85 | 33.80 ±0.29 | 33.54 ±0.20 | 33.74 ±0.26 | 33.71 ±0.24 |
| 13 | (1, 4, 7, 2) | 42.96 ±9.15 | **14.47** ±2.40 | 59.22 ±12.54 | **19.09** ±3.84 | 33.70 ±0.34 | 33.75 ±0.22 | 33.63 ±0.21 | 33.67 ±0.28 |
| 14 | (1, 3, 8, 2) | 65.32 ±13.63 | **35.25** ±5.75 | 93.44 ±13.87 | **51.72** ±8.90 | 38.30 ±0.26 | 38.39 ±0.24 | 38.28 ±0.27 | 38.36 ±0.35 |
| 15 | (1, 4, 8, 2) | 113.36 ±17.49 | **44.60** ±6.77 | 159.83 ±25.30 | **64.87** ±7.88 | 38.43 ±0.17 | 38.18 ±0.22 | 38.33 ±0.29 | 38.24 ±0.36 |
| 16 | (1, 3, 9, 2) | 156.88 ±23.55 | **86.26** ±16.11 | 222.42 ±43.47 | **122.44** ±21.53 | 43.02 ±0.40 | 42.98 ±0.45 | 42.87 ±0.26 | 43.11 ±0.29 |
| 17 | (1, 4, 9, 2) | 260.78 ±49.61 | **95.92** ±16.68 | 345.68 ±38.97 | **142.71** ±27.75 | 43.05 ±0.19 | 42.94 ±0.31 | 42.98 ±0.36 | 43.00 ±0.24 |
| 18 | (1, 3, 10, 2) | 367.77 ±71.46 | **199.80** ±35.50 | 555.65 ±121.90 | **337.10** ±52.84 | 47.65 ±0.34 | 47.36 ±0.16 | 47.62 ±0.30 | 47.67 ±0.42 |
| 19 | (1, 4, 10, 2) | 599.74 ±60.23 | **223.38** ±26.79 | 848.57 ±141.78 | **345.02** ±53.64 | 47.85 ±0.39 | 47.78 ±0.27 | 47.49 ±0.28 | 47.70 ±0.19 |
| 20 | (1, 3, 11, 2) | 914.39 ±217.86 | **515.34** ±85.61 | 1312.81 ±302.78 | **800.94** ±169.25 | 52.17 ±0.22 | 52.58 ±0.35 | 52.29 ±0.32 | 52.21 ±0.29 |
| 21 | (1, 4, 11, 2) | 1472.73 ±254.76 | **543.15** ±97.44 | 2168.73 ±450.03 | **819.44** ±159.47 | 52.36 ±0.27 | 52.35 ±0.41 | 52.37 ±0.46 | 52.26 ±0.30 |
| 22 | (1, 3, 12, 2) | 2336.52 ±492.69 | **1195.16** ±213.55 | 3171.35 ±554.37 | **1885.02** ±375.15 | 57.00 ±0.35 | 56.94 ±0.26 | 56.85 ±0.36 | 57.12 ±0.62 |
| 23 | (1, 4, 12, 2) | 3593.42 ±487.19 | **1385.19** ±187.00 | 5196.66 ±1063.94 | **2224.24** ±351.60 | 56.93 ±0.24 | 56.94 ±0.24 | 56.80 ±0.27 | 56.94 ±0.35 |
| 24 | (1, 3, 13, 2) | 5366.69 ±807.68 | **2721.86** ±337.95 | 6933.16 ±432.32 | **4505.52** ±811.02 | 62.26 ±1.69 | 61.63 ±0.39 | 63.83 ±2.43 | 61.67 ±0.29 |
| 25 | (1, 4, 13, 2) | 6997.37 ±338.99 | **3349.16** ±517.48 | 7200.00 ±0.00 | **4710.61** ±855.90 | 69.72 ±11.30 | **61.60** ±0.46 | 75.14 ±13.01 | **61.83** ±0.37 |

Table 4: Our test setup for the Schedule($C, p$) domain (lower values better). ID refers to the problem ID in the test set. $\theta$ refers to the parameters passed to the problem generator for generating the problem. Times indicate the seconds required to find a policy. Similarly, costs are reported as average costs obtained by executing the computed policy for 100 trials. We ran our experiments using a different random seed for 10 different runs and report average metrics up to one standard deviation. Better metrics are at least 5% better and are indicated using bold font.

| ID | $\theta$ | Time($x \equiv$ LRTDP) $x$ | Ours + $x$ | Time($x \equiv$ Soft-FLARES) $x$ | Ours + $x$ | Cost($x \equiv$ LRTDP) $x$ | Ours + $x$ | Cost($x \equiv$ Soft-FLARES) $x$ | Ours + $x$ |
|---|---|---|---|---|---|---|---|---|---|
| 0 | (1, 2) | 0.02 ±0.01 | 0.02 ±0.01 | 0.02 ±0.01 | 0.02 ±0.01 | 6.41 ±0.13 | 6.32 ±0.13 | 6.35 ±0.08 | 6.45 ±0.15 |
| 1 | (1, 3) | 0.08 ±0.02 | **0.06** ±0.02 | 0.07 ±0.02 | 0.07 ±0.02 | 9.53 ±0.12 | 9.50 ±0.09 | 9.60 ±0.13 | 9.52 ±0.12 |
| 2 | (1, 4) | 0.32 ±0.05 | **0.16** ±0.03 | 0.33 ±0.05 | **0.19** ±0.03 | 12.72 ±0.14 | 12.73 ±0.14 | 12.81 ±0.15 | 12.77 ±0.15 |
| 3 | (1, 5) | 1.58 ±0.25 | **0.41** ±0.06 | 1.60 ±0.34 | **0.46** ±0.09 | 16.00 ±0.14 | 15.97 ±0.17 | 15.89 ±0.11 | 15.95 ±0.19 |
| 4 | (1, 6) | 6.45 ±0.77 | **1.02** ±0.22 | 7.36 ±1.28 | **1.21** ±0.20 | 19.06 ±0.16 | 19.16 ±0.16 | 19.17 ±0.24 | 19.14 ±0.14 |
| 5 | (1, 7) | 36.46 ±7.19 | **2.46** ±0.56 | 35.61 ±6.35 | **3.10** ±0.59 | 22.37 ±0.19 | 22.45 ±0.19 | 22.45 ±0.25 | 22.28 ±0.24 |
| 6 | (1, 8) | 145.33 ±24.97 | **6.58** ±1.25 | 142.70 ±18.86 | **8.42** ±1.93 | 25.57 ±0.18 | 25.56 ±0.09 | 25.52 ±0.23 | 25.59 ±0.26 |
| 7 | (1, 9) | 616.36 ±140.89 | **14.92** ±1.65 | 622.93 ±61.96 | **19.85** ±3.00 | 28.78 ±0.18 | 28.61 ±0.16 | 28.73 ±0.25 | 28.88 ±0.18 |
| 8 | (1, 10) | 3036.01 ±507.41 | **38.89** ±7.41 | 2662.96 ±361.97 | **48.08** ±10.54 | 31.95 ±0.08 | 31.95 ±0.23 | 31.96 ±0.20 | 32.01 ±0.21 |
| 9 | (1, 11) | 7200.00 ±0.00 | **85.99** ±9.94 | 7200.00 ±0.00 | **122.28** ±19.92 | 81.68 ±16.65 | **35.06** ±0.29 | 74.56 ±19.98 | **35.10** ±0.28 |
| 10 | (1, 12) | 7200.00 ±0.00 | **220.64** ±43.34 | 7200.00 ±0.00 | **313.88** ±48.81 | 87.85 ±11.06 | **38.26** ±0.23 | 81.42 ±16.54 | **38.20** ±0.25 |
| 11 | (1, 13) | 7200.00 ±0.00 | **490.90** ±90.83 | 7200.00 ±0.00 | **692.75** ±126.60 | 89.45 ±12.88 | **41.50** ±0.40 | 89.54 ±11.68 | **41.43** ±0.22 |
| 12 | (1, 14) | 7200.00 ±0.00 | **1153.32** ±117.37 | 7200.00 ±0.00 | **1710.56** ±278.99 | 87.26 ±12.32 | **44.69** ±0.13 | 93.66 ±10.10 | **44.74** ±0.27 |
| 13 | (1, 15) | 7200.00 ±0.00 | **2514.16** ±337.59 | 7200.00 ±0.00 | **3639.92** ±627.08 | 86.84 ±11.41 | **47.77** ±0.17 | 85.14 ±13.60 | **47.74** ±0.34 |

Table 5: Our test setup for the Delicate Can($c$) domain (lower values better). ID refers to the problem ID in the test set. $\theta$ refers to the parameters passed to the problem generator for generating the problem. Times indicate the seconds required to find a policy. Similarly, costs are reported as average costs obtained by executing the computed policy for 100 trials. We ran our experiments using a different random seed for 10 different runs and report average metrics up to one standard deviation. Better metrics are at least 5% better and are indicated using bold font.

| ID | $\theta$ | Time($x \equiv$ LRTDP) $x$ | Ours + $x$ | Time($x \equiv$ Soft-FLARES) $x$ | Ours + $x$ | Cost($x \equiv$ LRTDP) $x$ | Ours + $x$ | Cost($x \equiv$ Soft-FLARES) $x$ | Ours + $x$ |
|---|---|---|---|---|---|---|---|---|---|
| 0 | (2) | 0.01 ±0.01 | 0.01 ±0.00 | **0.00** ±0.00 | 0.01 ±0.01 | 5.40 ±0.07 | 5.40 ±0.08 | 5.45 ±0.07 | 5.40 ±0.10 |
| 1 | (3) | 0.03 ±0.01 | **0.02** ±0.01 | 0.03 ±0.01 | **0.02** ±0.01 | 7.42 ±0.12 | 7.50 ±0.15 | 7.53 ±0.09 | 7.48 ±0.06 |
| 2 | (4) | 0.11 ±0.03 | **0.07** ±0.02 | 0.12 ±0.03 | **0.09** ±0.02 | 9.58 ±0.11 | 9.53 ±0.16 | 9.56 ±0.14 | 9.62 ±0.09 |
| 3 | (5) | 0.49 ±0.08 | **0.22** ±0.05 | 0.51 ±0.08 | **0.40** ±0.06 | 11.71 ±0.20 | 11.72 ±0.12 | **12.48** ±0.69 | 14.22 ±2.49 |
| 4 | (6) | 2.56 ±0.55 | **0.80** ±0.13 | 2.14 ±0.26 | **1.79** ±0.23 | 13.82 ±0.15 | 13.75 ±0.07 | **15.02** ±1.09 | 25.16 ±4.14 |
| 5 | (7) | 12.55 ±2.20 | **3.25** ±0.56 | 8.08 ±0.92 | **6.81** ±1.41 | 15.94 ±0.09 | 15.86 ±0.08 | **18.87** ±1.44 | 23.32 ±4.09 |
| 6 | (8) | 54.31 ±11.04 | **14.12** ±2.60 | 31.86 ±5.15 | **27.65** ±4.41 | 18.08 ±0.13 | 18.07 ±0.14 | 22.16 ±1.68 | 22.01 ±1.52 |
| 7 | (9) | 244.32 ±47.46 | **54.60** ±10.68 | 101.63 ±20.29 | **90.57** ±15.57 | 20.04 ±0.13 | 20.15 ±0.14 | 25.18 ±2.65 | **22.70** ±1.22 |
| 8 | (10) | 960.96 ±105.78 | **188.50** ±29.21 | 331.06 ±49.98 | **293.54** ±36.06 | 22.28 ±0.17 | 22.17 ±0.13 | 28.54 ±2.09 | **26.29** ±1.73 |
| 9 | (11) | 4691.23 ±859.08 | **698.09** ±143.63 | 990.65 ±144.48 | **983.51** ±144.08 | 24.30 ±0.12 | 24.41 ±0.17 | 30.05 ±1.37 | **28.02** ±1.40 |
| 10 | (12) | 7200.00 ±0.00 | **3769.57** ±439.21 | 3347.02 ±540.66 | **2911.30** ±397.41 | 27.46 ±0.83 | 26.46 ±0.14 | 32.23 ±2.35 | **29.49** ±1.47 |
| 11 | (13) | 6980.00 ±660.00 | 7183.58 ±49.27 | 6980.48 ±658.56 | 7132.65 ±202.04 | 96.62 ±10.13 | 96.12 ±11.65 | 96.50 ±10.49 | 96.48 ±10.56 |

Table 6: Our test setup for the Gripper($b$) domain (lower values better). ID refers to the problem ID in the test set. $\theta$ refers to the parameters passed to the problem generator for generating the problem. Times indicate the seconds required to find a policy. Similarly, costs are reported as average costs obtained by executing the computed policy for 100 trials. We ran our experiments using a different random seed for 10 different runs and report average metrics up to one standard deviation. Better metrics are at least 5% better and are indicated using bold font.

| ID | $\theta$ | Time($x \equiv$ LRTDP) | | Time($x \equiv$ Soft-FLARES) | | Cost($x \equiv$ LRTDP) | | Cost($x \equiv$ Soft-FLARES) | |
|---|---|---|---|---|---|---|---|---|---|
| | | $x$ | Ours + $x$ | $x$ | Ours + $x$ | $x$ | Ours + $x$ | $x$ | Ours + $x$ |
| 0 | (1) | 0.01 ±0.01 | **0.00** ±0.01 | **0.00** ±0.00 | 0.01 ±0.01 | 3.25 ±0.06 | 3.25 ±0.05 | 3.23 ±0.06 | 3.25 ±0.04 |
| 1 | (2) | 0.02 ±0.01 | 0.02 ±0.01 | 0.02 ±0.01 | 0.02 ±0.01 | 5.48 ±0.08 | 5.53 ±0.08 | 5.51 ±0.10 | 5.55 ±0.07 |
| 2 | (3) | **0.10** ±0.02 | 0.12 ±0.03 | 0.12 ±0.03 | 0.12 ±0.03 | 9.71 ±0.07 | 9.76 ±0.10 | 9.76 ±0.07 | 9.75 ±0.09 |
| 3 | (4) | 0.33 ±0.05 | 0.34 ±0.05 | **0.30** ±0.06 | 0.38 ±0.07 | 11.99 ±0.10 | 12.03 ±0.16 | 12.00 ±0.08 | 12.03 ±0.10 |
| 4 | (5) | **1.36** ±0.23 | 1.62 ±0.28 | **1.82** ±0.35 | 2.21 ±0.47 | 16.27 ±0.09 | 16.23 ±0.12 | 16.22 ±0.06 | 16.26 ±0.15 |
| 5 | (6) | 3.38 ±0.39 | 5.17 ±1.65 | **4.64** ±0.80 | 5.16 ±0.64 | 18.51 ±0.21 | 18.54 ±0.14 | 18.53 ±0.12 | 18.44 ±0.07 |
| 6 | (7) | **14.00** ±2.83 | 17.67 ±3.43 | **18.07** ±2.13 | 22.28 ±4.48 | 22.82 ±0.08 | 22.78 ±0.10 | 22.80 ±0.19 | 22.73 ±0.15 |
| 7 | (8) | **34.11** ±7.52 | 38.35 ±4.68 | **43.88** ±8.90 | 55.78 ±12.25 | 24.96 ±0.12 | 24.99 ±0.13 | 24.96 ±0.14 | 24.95 ±0.14 |
| 8 | (9) | **108.94** ±19.36 | 133.06 ±24.75 | **175.48** ±33.68 | 186.91 ±24.64 | 29.17 ±0.10 | 29.26 ±0.20 | 29.23 ±0.14 | 29.08 ±0.17 |
| 9 | (10) | 264.59 ±28.50 | 362.16 ±60.65 | **392.91** ±69.13 | 463.30 ±78.81 | 31.48 ±0.20 | 31.54 ±0.15 | 31.47 ±0.14 | 31.54 ±0.17 |
| 10 | (11) | **867.91** ±195.44 | 1028.70 ±172.62 | **1256.52** ±277.94 | 1509.25 ±195.14 | 35.78 ±0.09 | 35.73 ±0.15 | 35.70 ±0.15 | 35.84 ±0.16 |
| 11 | (12) | **2037.65** ±326.63 | 2454.41 ±377.38 | **2941.24** ±457.95 | 3461.67 ±595.74 | 38.07 ±0.13 | 37.98 ±0.15 | 38.03 ±0.18 | 38.10 ±0.12 |