# OpenReview forum: "Learning Generalized Policy Automata for Relational Stochastic Shortest Path Problems"
_NeurIPS.cc/2022/Conference — NeurIPS 2022 Accept_

### Official Review · Reviewer_qpA1 · 2022-06-24

**Rating:** 7
**Confidence:** 3
**Soundness:** 3 good
**Presentation:** 3 good
**Contribution:** 3 good

**Summary:**

In this paper, the authors propose an approach to learning Generalized Policy Automata (GPA) so as to accelerate the solution process of Stochastic Shortest Path Problems (SSP). GPAs are a non-deterministic finite-state automata that compactly encodes generalized knowledge. Furthermore, GPA imposes hierarchical constraints on the state space of the SSP and prunes the considered action space. To learn GPA, they group different objects through a canonical abstraction, and then iteratively compute GPA from a small training set containing solutions like SSP instances. Finally, they theoretically proved that the method has completeness and hierarchical optimality. Experimental results also verify that the method significantly outperforms state-of-the-art SSP solvers.

**Questions:**

(1) You mentioned in the Solving SSPs using GPAs section that there is no abstract translation in general policy automata (line 228). When does this phenomenon happen? In addition, if it happens, does it cause the SSP solver to have no speedup or even less efficient?

(2) In the experimental section, you only used a small training set when learning GPA, and finally got very good results. If you continue to increase the number of training sets, will the experimental results gain a significant improvement?

(3) As I mentioned earlier, this paper can be better comprehensive by adding some explanatory pictures for some complex contents (such as the section on "Canonical Abstraction") and illustrating the statement of experimental standards in priority.

**Limitations:**

In my opinion, the authors fail to mention some limitations of their approach in the article, which may overlook some potential problems. As the authors mention in the section "Solving SSP with GPA", there is no abstract translation in the general policy automata (line 228). If this happens frequently, it may affect the effectiveness of their approach. Therefore, I think this is a limitation worthy of further consideration.

**Strengths And Weaknesses:**

Originality：
(1) Compared with other methods, the method of this paper can automatically construct the GPA without any human intervention. In addition, canonical abstractions are used to extract problem-specific features such as object names and object counts.
(2) Another key difference from other techniques in existence is that the authors' method can easily incorporate solutions from new examples into the controller without having to remember any earlier examples, leading to better scalability of GPA learning.

Quality：
(1) I find that the overall structure of the paper is good, that leverages clear method structures, exhibits comprehensive comparison to related work and builds up reasonable comparative experiments.
(2) As I said above, the overall structure of this paper is good. However, the position where the evaluation criteria appear is not very appropriate. They should be introduced at the beginning of the experimental section, which will make the whole experimental process more reasonable.

Clarity:
(1) The overall meaning of this paper is transparent, most of the methods are well explained, and the theorems are clearly proved.
(2) Although the paper contains examples, but there is only a lengthy description in the section “Canonical Abstraction”. The work would benefit by including some clear, illustrative pictures into the main body of the paper, to better motivate and illustrate the work.

Significance：
In my opinion, this paper addresses an interesting research topic where they exploit knowledge from solutions of SSP instances with small object counts to learn GPAs that allow effective pruning of the search space of related SSPs with larger object counts. This ability is essential for solving stochastic shortest path problems because of its excellent generalization ability and solution efficiency.

---

> ### Author Response · Authors · 2022-08-01
> **Thank you for your feedback. We address your questions below.**
>
> Thank you for the feedback and comments. We appreciate your points on the presentation and have uploaded an updated version of the supplementary material that includes some figures and fleshed-out examples to help improve the clarity of the paper (Appendix C). If accepted, we also plan to purchase additional pages to incorporate those changes in the final version as well as incorporate the minor suggestions that you have made (organization of empirical evaluation, etc.).
>
> **Q1.** The exclusion of transitions that have no abstract translations (line 228/lines 4-5 of Alg.1) prevents the algorithm from exploring actions that are unlikely to be useful based on the training data. This is the key insight into our method. Alg.1 penalizes such actions by setting their costs to infinity. Our experiments show that this leads to computational efficiency.
>
> Naturally, this phenomenon can also occur in such a way that optimal actions of a problem can also get pruned out. This typically happens when the training data is not sufficient to generate a GPA that can generalize well to larger problems. In this case, you are correct that the savings from our approach will be reduced. However, empirically, we have observed that learning-from-small-examples is an effective generalization approach. We provide an intuitive answer in our response to Q2 below.
>
> **Q2.** We experimented with increasing the training set size and found that the GPA remained the same after a few small problems. For example, for Keva problems, where the objective was to build a tower of height “h”. If the training set contained towers of up to height 6, the GPA formed generalizes to towers of any height. This is because tower height 1 (the base) and tower heights 2-6 (the remaining floors) are special cases that require some specific transitions using canonical abstraction. Intuitively, we expect that a good policy for tower height 6 could be iteratively used to construct larger towers using the same patterns of plank arrangements. In this case, we wouldn't expect larger problems to help improve the generalized controller. Empirically, we observed that adding solutions of towers with a height of more than 6 to the training set did not improve the controller further.
>
> Automatically finding the smallest set of training problems that can enable good generalization is an interesting but non-trivial research problem. Empirically, research in this area has shown that small examples can be quite effective in constructing a good generalization (Bonet et al. 2009; Bonet et al. 2018; Segovia-Aguas et al. 2020).
>
> **Q3.** Thank you for your feedback. We have incorporated better descriptions and illustrative examples on canonical abstraction for improved clarity in the updated supplementary material (Appendix C). For the final version, we also plan to purchase additional pages to further improve the clarity by incorporating the other suggestions that you have made.
>
> **References**
>
> Bonet, B., Palacios, H. and Geffner, H., 2009. Automatic derivation of memoryless policies and finite-state controllers using classical planners. In Proc. ICAPS.
>
> Bonet, B. and Geffner, H., 2018. Features, projections, and representation change for generalized planning. In Proc. IJCAI.
>
> Segovia-Aguas, J., Jiménez, S. and Jonsson, A., 2020. Generalized planning with positive and negative examples. In Proc. AAAI.

---

> > ### Comment · Reviewer_qpA1 · 2022-08-09
> > **After reading response.**
> >
> > Thanks for the thorough response!
> >
> > Learning-from-small-examples: The examples you use clearly illustrate the situations in which you can achieve good experimental results by training with only a small sample dataset. Although research in this area has shown that small samples can be very effective in establishing good generalization. However, it is worth exploring whether some special domains are suitable for this kind of " Learning-from-small-examples" situation. In other words, what kind of structured problem fits this solution situation?
> >
> > Find the Suitable Dataset: It seems to me that your approach needs to pick as small and suitable dataset as possible to produce good generalization. As you mentioned, automatically finding the smallest set of training problems that can achieve good generalization is an interesting and non-trivial job. My feeling is that this loss of information about choosing the right dataset by experience or something should be avoidable.
> >
> > Clarifying Illustrations: The new illustrations and descriptions in Appendices C seem to significantly strengthen the paper.

---

> > > ### Author Response · Authors · 2022-08-09
> > > **Post-response**
> > >
> > > Thank you for your feedback and constructive comments. We are glad that the illustrations helped improve the clarity of the paper.

---

### Official Review · Reviewer_4Xab · 2022-07-10

**Rating:** 7
**Confidence:** 3
**Soundness:** 3 good
**Presentation:** 3 good
**Contribution:** 2 fair

**Summary:**

The paper introduces the concept of generalized policy automata (GPA) to speed up the solution of stochastic shortest path problems (SSPs), defined here within a more general AI Planning perspective. The intuition is to leverage GPAs within a "learn-from-small-examples" framework, i.e., they capture the (abstract) states and optimal controls of small instances (the "training" set), and are subsequently used in a GPA-accelerated SSP solution (specifically, by adjusting the SSP cost function accordingly). Numerical results evaluate the approach on classical planning benchmarks.

**Questions:**

1. Is it possible to incorporate more examples of the type of learning that is being performed by the approach?

2. Do the authors have any (theoretical) intuition on when learning from small examples in this particular context is effective? What is the fundamental structure of the smaller instances that is preserved in larger problems for this approach to be successful?

3. Are there any gains if more time and instances are allotted to the training set? Is there any risk, for example, of an "oversampling" in this context? (I believe not because of Theorem 3.3?)



**Strengths And Weaknesses:**

Strengths
- Innovative structure to capture knowledge associated with optimal controls
- Good computational results for the domains tested

Weaknesses
- In my view, limited analysis or discussion of why and when the method is effective
- Numerical results seem quite restricted

Major Comments

In general, I greatly appreciated the GPA concept; I found it to be innovative and overall intuitive to understand, as essentially it projects the original system to a smaller (abstract) set of states and actions. The results for the four planning domains tested are also promising, especially when evaluating the results for Keva and Rover. However, I have two major concerns associated with the methodology and the numerical study.

(i) [Methodology.] I believe my major question, perhaps due to some misunderstanding on my part, is fundamentally "why would this method work?" More precisely, there is little intuition presented on when the small instances have enough information to capture good controls, or why that should always the case. What are the underlying properties of SSPs that motivate heuristics based on small optimal portions, and when (or if) would this break? Could it be the case that this information is "deceiving," in that larger problem instances have significantly different controls (e.g., Gripper)?

While I understand that the analysis is a more computational proof of concept, I felt that any intuition, such as through more informative examples, would be quite effective in demonstrating why the approach is effective.

(ii) [Numerically.] Related to point (i), the numerical study essentially focuses on a horse-race comparison between approaches in terms of time and solution cost. However, I believe that for such a new idea there are other metrics that could be as important as the total runtime for drawing insights. For example, the training setup is rather limited (only ten instances!) - is there any gains in increasing the learning period? If not, why would that be the case? Furthermore, are there any properties of states that the policy was more "accurate" with respect to the optimal control?

Overall, I enjoyed the idea but feel that a more extensive numerical analysis is critical to this work, again to shine light into when the method could be most effective.

---

> ### Author Response · Authors · 2022-08-01
> **Thank you for your feedback. We address your questions below.**
>
> Thank you for your feedback and questions.
>
> **Numerical Study**
>
> The performance of our method is in fact invariant w.r.t. other metrics such as learning time, batch sizes, etc. with the same training data as our approach will yield the same GPA irrespective of the batches/orders in which the training data is presented. This is one of the strengths of our approach: in contrast to Deep Learning, we use abstraction to learn the GPA controller from training data, and increasing the total learning time allotted would not affect performance. Our training time was ~10 seconds for each of the domains (lines 315-316). In addition to the strong theoretical guarantees, our approach also provides strong few-shot learning. We welcome suggestions for additional metrics.
>
> We now discuss the specific questions you raised below.
>
> **Q1.** Thank you for your feedback! We have uploaded a revised version of the supplementary material that improves the clarity of the paper and includes illustrative figures that better explain the type of learning that our approach is performing (Appendix C). If accepted, we also plan to purchase an additional page to incorporate these additions and further expand on some of the formally dense parts of the paper.
>
> **Q2.** The intuition behind our approach is to use an auto-generated abstraction to distill the knowledge contained in solutions to small instances of a problem, and to use that knowledge to aid the solution process for larger problems. Prior work in this area shows that some of the common structures can be iterative constructs and situations where similar sequences of abstract actions apply. E.g., the Keva task requires a complex arrangement of planks, which occurs at each level of the tower. The presented approach automatically discovers and effectively transfers such patterns in solutions even when tower heights are changed far beyond training data.
>
> This intuition builds on related work on lifted inference in probabilistic reasoning.  Earlier work in sequential decision-making considered the related idea of lifted computation for exact generalized solutions (e.g. Boutilier et al. 2001; Sanner et al. 2010; Wang et al. 2010). However, these approaches suffered from complications similar to shattering in lifted inference (Kisynski et al. 2009).
>
> Our work presents the first broadly successful instantiation of the concept to SSPs by building a non-deterministic, lifted controller using auto-generated relational formulas as features. It differs from the previous approaches on lifted sequential decision-making cited above in that we learn and use a generalized controller to guide the solution process for new problem instances rather than attempting to compute exact, lifted solutions for all possible problem instances. As a result, we are able to transfer learned knowledge more broadly without suffering from shattering-related issues.
>
> As in almost all learning literature, if the test instances come from a significantly different distribution than the training instances then the performance of our approach is likely to suffer.
>
> **Q3.** You are correct in that Thm. 3.3 guarantees that the presented algorithm will not suffer due to oversampling or overtraining. We use every available transition in the training set and make only a single pass over the training data to compute the GPA.
> This is a valuable advantage of the proposed approach. In addition, our approach does not suffer from catastrophic forgetting. Our method does not require hyperparameter tuning and our training time is a fraction of the time required in some DL models (lines 315-316) and works in few-shot settings. We believe that this approach can play a role complementary to DL pipelines, perhaps also for efficiently generating training data for a DL pipeline.
>
> Our approach, to the best of our knowledge, is one of the first approaches to provide crisp theoretical guarantees of finding a hierarchically optimal solution given a fixed training set and also generalize to larger problems.
>
> **References**
>
> Boutilier, C., Reiter, R. and Price, B., 2001. Symbolic dynamic programming for first-order MDPs. In IJCAI.
>
> Sanner, S. and Kersting, K., 2010. Symbolic dynamic programming for first-order POMDPs. In Proc. AAAI.
>
> Wang, C. and Khardon, R., 2010, July. Relational partially observable MDPs. In Proc. AAAI.
>
> Kisynski, J. and Poole, D. 2009. Constraint processing in lifted probabilistic inference. In Proc. UAI.

---

> > ### Comment · Reviewer_4Xab · 2022-08-09
> > **Feedback**
> >
> > Thank you for the very detailed answer and the clarifying illustrations in the appendix (they are very insightful). You also addressed all my major concerns and misunderstandings. I will update my score accordingly.

---

> > > ### Author Response · Authors · 2022-08-09
> > > **Post-response**
> > >
> > > Thank you for your review. We appreciate the comments pertaining to improving the clarity of our paper and are happy that the updated appendix and discussion helped in addressing your concerns.

---

### Official Review · Reviewer_qjGU · 2022-07-11

**Rating:** 8
**Confidence:** 4
**Soundness:** 4 excellent
**Presentation:** 3 good
**Contribution:** 3 good

**Summary:**

The paper introduces a learning algorithm for a class or relational stochastic shortest path problems. The algorithms specialized in relational domains where states have meaningful factors out of named relationships -called predicates– and objects. Using previous ideas for generalized planning, the algorithm defines a new cost function for the original problem that assigns infinite cost to transitions not properly rendered in the abstraction. Such a problem is used to obtain a policy using an existing algorithm. If that policy is proper, that could be returned. Otherwise, the values of the previous algorithms can be used to solve the problem with the original cost.
In summary, the algorithm synthesizes a generalized policy automata (GPA) and attempts a solution to the global problem using its cost.

**Questions:**

1. How can this be used in domains where there are no unary predicates? For instance, suppose a graph that has proportional in the edges for not in the nodes. Each proposition can be seen as a binary predicate over the notes.
2. In the experiments,
	2.1. How frequently was the policy using the GPA proper?
	2.2. In that case, how much is gained by running the full algorithm?
	2.3. In the domains, how many costs changed to $\infty$ because of the GPA
3. Please, elaborate on the relationship between the submission and "Learning generalized relational heuristic networks for model-agnostic planning. In AAAI, 2021.?
4. In problems with the same GPA, could policy could be used directly?


**Limitations:**

I think the paper comments enough about the technical limitations. This is a well-defined class of problems, and their solutions are strictly algorithmic with predictable properties.

**Strengths And Weaknesses:**

The strongest points of the paper are:
- Use of abstraction previously used for deterministic domains.
- Clear characterization of what the generalization means.
- Low-data requirements
- Clearly significant empirical results

Weakness
- The paper is built around the assumption that what matters is the accuracy in comparison with other algorithms that attach the problem directly. That leads to not-so-surprising statements like theorem 3.2, as its proof applies to any algorithm that generates a candidate of the solution, returns it if it works, and falls back to a more complex solution.
- The notion of canonical abstraction is not explained enough to understand the scope of the technique. The example right before 3.3 offers some ideas but I would not expect users unfamiliar with the previous work on generalized planning to understand that. The domains would have presented an opportunity to describe what the GPA is doing in those cases.
- The canonical abstraction depends strongly on the relational domain.

In my opinion, the strengths outweigh the weakness. The paper is presented as an empirical algorithm, so it makes sense to focus the evaluation on existing tasks. I would have say that understanding the expressivity of the GPA is important and beyond the scope of the paper.

Some additional comments:
- please revise this paragraph: *Highlight [page 4]:* Given a predefined set of abstraction predicates, canonical abstractions group together different objects based on the subset of abstraction predicates that they satisfy in a state. Each subset of abstraction predicates is known as a summary element
- Why? *Highlight [page 4]:* An object o belongs to exactly one summary element in any given concrete state s and this is the maximal summary element for that object.

---

> ### Author Response · Authors · 2022-08-01
> **Thank you for your feedback. We address your questions below.**
>
> Thank you for your comments and questions. We appreciate your suggestions for presentation and have included detailed illustrative examples in the updated supplementary material (Appendix C). We plan to purchase additional pages to incorporate them including suggestions for improved text in Sec. 3.2 and other minor suggestions in the final version.
>
> **Q1.** You are right that canonical abstraction is a function of the relational domain. However, it does allow us to apply learned knowledge on problem instances that are much larger than what we saw in training data.
>
> Domains that do not have any unary predicates reduce the expressivity when using canonical abstraction. We chose Canonical Abstractions since they provide an easy way to generate good features without any human input. However, as mentioned in lines 401-404, our approach can use any input abstraction mechanism to generate the GPA. Recently, description logic based methods have been proposed that could provide another source of such abstractions (Frances et al. 2021; Drexler et al. 2022; Ferber et al. 2022). Although the problem of generating a reduced feature set starting from description logic syntax has not been adequately explored for stochastic settings, this is a promising direction for future work.
>
> **Q2.1** Except for a few problems in Delicate Can (where our approach timed out), we found that the policies found by using the GPA were already “partial proper” and were similar in cost when compared to the baselines. We will mention this in the paper.
>
> **Q2.2** Our results show that our approach results in significant computational speedups even when the returned policy (line 11 of Alg. 1), is “partial proper” as defined in Sec. 2 (lines 104-109).
> If one were to include the complete algorithm with the bootstrapped values (lines 13-15), we would expect similar savings. In fact, if the policy returned by line 11 is already optimal, then lines 13-15 would return much faster when using a good heuristic since the costs of the “new” transitions would be greater than the converged values. If a blind heuristic is used instead, lines 13-15 would require computations of all the “new” transitions until they are greater than that of the policy computed in line 11.
>
> **Q2.3** Due to the way we implemented our source code, we do not have exact metrics for counting the total transitions discarded. The metrics we collected were the total number of bellman updates performed and states explored in converging to a solution. We believe this is also informative since it indicates the total computational effort saved as a part of removing such transitions. We provide some examples below for cases where both the baselines were able to solve the problems.
> ```
> Problem: Bellman updates: our approach/baseline | Time taken: our approach/baseline
> Rover ID 24: 18.5M/41.7M | 50m/96m
> Schedule ID 8: 193k/10M | 42s/50m
> Gripper ID 11: 7.7M/7.7M | 2600s/2100s
> ```
> We believe the additional time in Gripper ID 11 was due to the overhead introduced by abstraction since the computational effort and even the state space explored (376k states) were similar. This is discussed in lines 337-344 of the paper.
>
> **Q3** The work by Karia et al. (2021) is a neuro-symbolic approach for synthesizing generalized heuristics for deterministic classical planning problems and does not address stochasticity. On the other hand, our method learns a controller and transfers it to optimize bellman backups in SSPs. We will mention this in the final version.
>
> **Q4** The GPA can indeed be used directly as follows: “select any concrete action that matches this abstract state and action, and transition to a new state in the GPA”, however, this has the limitation that this induced reactive policy is non-deterministic: many grounded versions of abstract actions are possible, and without the rest of our algorithm, it will be difficult to choose the "best" one. As a result, using the GPA directly as a non-deterministic reactive policy will not have the guarantees of hierarchical optimality that our algorithm maintains.
>
> **References**
>
> Frances, G., Bonet, B. and Geffner, H., 2021. Learning general planning policies from small examples without supervision. In Proc.  AAAI.
>
> Karia, R. and Srivastava, S., 2021. Learning generalized relational heuristic networks for model-agnostic planning. In Proc AAAI.
>
> Drexler, D., Seipp, J. and Geffner, H., 2022. Learning Sketches for Decomposing Planning Problems into Subproblems of Bounded Width. In Proc. ICAPS.
>
> Ferber, P., Cohen, L., Seipp, J. and Keller, T., 2022. Learning and Exploiting Progress States in Greedy Best-First Search. In Proc. IJCAI.

---

### Official Review · Reviewer_3WfX · 2022-07-11

**Rating:** 7
**Confidence:** 5
**Soundness:** 4 excellent
**Presentation:** 3 good
**Contribution:** 2 fair

**Summary:**

This work presents an approach for learning state-action abstraction in stochastic shortest path problems (SSPs). The paper introduces  Generalized Policy Automata (GPA), which are hypergraphs wherein edges representing abstract actions connect a start abstract state to some set of destination abstract states. Given a planning domain described in a symbolic description language, the paper proposes to automatically learn a GPA on small instances of a given planning domain, and then use the learned GPA to accelerate planning with an off-the-shelf SSP solver in new, possibly larger, problem instances. This is done by pruning state transitions that don't correspond to any hyperedge in the learned GPA, thus reducing the amount of computation needed. If the pruned version of the problem cannot be solved, the algorithm resorts to planning with the original problem. The authors analyze some of the theoretical properties of the method. Finally, experimental results show that the proposed approach can accelerate state-of-the-art SSP solvers in a number of planning domains represented in the PPDDL symbolic language.

**Questions:**

Considering the discussion above, my main questions are:

- Can you motivate the choice of evaluation domains? How does your method perform in other SSP problems described in PPDDL that are commonly used (e.g., triangle tireworld or exploding blocksworld).

- How were the hyperparameters for Soft-FLARES chosen?

- Other recent approaches such as ASNets (Toyer et al., 2020), which is mentioned in the introduction, would also be relevant for this problem and not included in the evaluation. Can you elaborate on why this choice was made? Similarly, a natural baseline to include whenever pruning is used in PPDDL domains is a determinization-based approach, which ofter perform surprisingly well. The reference [1] above does this by learning a promising determinization in small problem instances, similarly to how the proposed approach does it. Is there a reason why this comparison is not applicable here?

**Limitations:**

The authors discuss several limitations of the approach in Section 6. I think it would also be important to include some discussion regarding the limitations I mentioned under "Significance" above.

**Strengths And Weaknesses:**

**Originality**

The proposed approach builds on a well-known notion of abstraction (canonical abstraction), but the proposed approach for learning and using GPAs is new, to the best of my knowledge. On the other hand, the idea of learning a sparse representation of planning problems in small instances and apply on larger instances has been explored before (see [1] below for an example), so the main novelty of this work lies in the use of GPA for this, and the completeness guarantees.

-----
[1] Pineda, L.E. and Zilberstein, S., 2014, May. Planning under uncertainty using reduced models: Revisiting determinization. In Twenty-Fourth International Conference on Automated Planning and Scheduling.

**Quality**

The proposed approach is sound, and the theoretical properties look correct to me. I have some concerns about the experiments. One is the choice of problems, which seems a bit arbitrary, considering that they are different than those used in other recent work, including the Soft-FLARES baseline, but also in ASNets by Toyer et al., 2020, and Karia Srivastava, 2021.
I also have some reservations about the hyperparameter tuning for Soft-FLARES. In the appendix it is mentioned that $t=4$ was selected, but this choice is not motivated. A look at the Soft-FLARES paper suggests that this parameter has strong impact on performance, and that policies of similar quality can be obtained faster with lower values of $t$, which would possibly decrease the perceived benefits of the GPA-based approach. Again, stronger motivation for this choice should be offered.

**Clarity**

The paper is reasonably well-written and the flow of ideas in the paper seems natural; the running example was also helpful. On the other hand, the paper is notation-heavy and somewhat dense, and it took a number of careful reads before I was confident that I understood the proposed approach. It would be helpful for readers to start with a high level overview of what the proposed technique is, without using any of the formal definitions, preferably including an image/diagram to illustrate how it works. I think the basic idea is intuitive enough that such an explanation is possible, but it's currently hidden under a dense layer of formalism.

**Significance**

I think the proposed approach is interesting and relevant to the probabilistic planning community. The experiments show that, when a symbolic model of the environment is available, the proposed approach can be beneficial in terms of solution quality and computation time. That being said, the applicability of this approach is limited to having a symbolic model of a problem available, which I'd argue it's often a harder problem than planning itself. Indeed, the statement in line 21 about human-engineering effort can also perfectly be applied to building a symbolic model, so I'd be wary to offer this as a strong motivation in favor of this approach. One advantage of deep-learning based approaches is precisely that it avoids human effort spend in hard-coding a good representation for the task at hand.

---

> ### Author Response · Authors · 2022-08-01
> **Thank you for your feedback. We address your questions below.**
>
> Thank you for the detailed feedback and suggestions for improving the presentation. We have included illustrative examples in the updated supplementary material (Appendix C). We will purchase additional pages to incorporate them along with other minor suggestions.
>
> **Symbolic Action Models**
>
> We agree with you that creating such models requires some human engineering. We would like to point out respectfully that the creation of simulators also requires extensive human engineering and most DRL methods use such simulator models due to the infeasibility of training a DRL agent from scratch directly in the real world. While both these approaches require some human engineering effort, we believe that they are complementary, advance the state of the art on both simulator-based and symbolic models, and would help in increasing the practical applicability of sequential decision making. For instance, the methods presented here show powerful sample efficiency and transfer characteristics.
>
> **Q1.** Following research on this topic (Toyer et al. 2018; Pineda et al. 2017; Pineda et al. 2019; etc.), we used a representative sample of commonly used problems from the IPCs augmented with a few additional challenging domains from related papers (Sec. 4, lines 276–278).
>
> Exploding blocksworld is not a GSSP since problems in this domain can have unavoidable deadends. As noted in lines 102-110, this paper focuses on GSSPs. Following your suggestion, we tried Triangle Tireworld (TT) instances from IPPC-08 during the rebuttal phase and obtained the following result (due to limited time and resources during the rebuttal phase, our training data generator, python based baselines, as well as GPA-versions, run out of memory for TT4 and larger):
>
> TT3:
> ```
> Algorithm: states explored/bellman updates/time taken
> GPA + LRTDP (Ours): 10.3k/16.4k/3.5s
> LRTDP: 16.6k/33k/2s
> ```
>
> Our approach reduces the computational effort, but it takes slightly more time, due to the overhead in abstract state computation (done once for every state).
>
> **Q2.**  We used hyper-parameters from the latest source code for Soft-FLARES (found [here](https://github.com/luisenp/mdp-lib/blob/master/test/planningServer.cpp#L190)). Our approach is less sensitive to hyper-parameter tuning because it embeds inside any given SSP solver and uses the same hyperparameter settings.
>
> We expect that decreasing t will not change our conclusions. t controls how states are labeled as solved by inspecting the *greedy* sub-tree from a state. In our case, states could get marked as deadends quickly irrespective of the horizon, whereas, this would be less frequent in the baseline case.
>
> To validate our hypothesis, we re-ran Soft-FLARES on Keva with t=2 and a time limit of 3600s. In each case, the costs obtained were similar. Examples of results [other instances were similar]:
> ```
> Problem ID....: time taken: GPA + Soft-FLARES(t=2)/Soft-FLARES(t=2)
> 09............: 374s/1470s
> 10............: 516s/2023s
> 11............: 668s/2510s
> 12............: 829s/2667s
> ```
> **Q3.** This paper addresses the problem of computing generalizable knowledge by learning generalized controllers that provide clear theoretical guarantees, while Toyer et al. (2019) address the problem of making NN receptive fields generalizable using additional heuristic inputs.
>
> Thank you for mentioning the related work by Pineda et al. (2014) [PZ14 henceforth]. Our approach differs significantly from that work. PZ14 builds sparse representations of SSP problems by reducing the branching factor in the “environment’s choices” (the set of probabilistic effects of an action), while our approach uses abstraction to create abstract controllers that generalize solutions to SSPs and reduces the branching factor in the agent’s choice (the set of applicable actions). Our approach always considers all possible outcomes of every action. We will include this in the paper.
>
> For evaluation, we chose a subsequent solver Soft-FLARES developed by the same team following their research advances because (a) they reported that Soft-FLARES outperforms determinization-based approaches (Pineda et al. 2019), and (b) subsequent work by Pineda et al. noted that finding good model reductions needed can be difficult (Pineda et al. 2017). Perhaps due to similar reasons, PZ14 was not used as an empirical baseline in the authors' follow-up work (Pineda et al. 2017).
>
> **References**
>
> Toyer, S., Trevizan, F., Thiébaux, S. and Xie, L., 2018. Action schema networks: Generalised policies with deep learning. In Proc. AAAI.
>
> Pineda, L. and Zilberstein, S., 2014, May. Planning under uncertainty using reduced models: Revisiting determinization. In Proc. ICAPS.
>
> Pineda, L., Wray, K. and Zilberstein, S., 2017. Fast SSP solvers using short-sighted labeling. In Proc. AAAI.
>
> Pineda, L. and Zilberstein, S., 2019. Soft labeling in stochastic shortest path problems. In Proc. AAMAS.

---

> > ### Comment · Reviewer_3WfX · 2022-08-09
> > **Thanks for the detailed response**
> >
> > I appreciate the discussion regarding the symbolic action models, and agree with the authors that both approaches are complementary. Likewise, thanks a lot for the clarifications regarding the experimental setup, and in particular for highlighting that your approach always considers all possible outcomes, in contrast with approaches using single outcome determinization. I still think it would be interesting to compare this method to some determinization-based approach, but I don't think this is a critical omission. After reading all reviews and the authors' response, I'll increase my score to Accept.

---

> > > ### Author Response · Authors · 2022-08-09
> > > **Post-response**
> > >
> > > Thank you for your thorough review. We are glad that the discussion addressed your major concerns.

---

### Meta-Review · Area_Chair_QUk3 · 2022-08-23

**Recommendation:** Accept
**Confidence:** Certain

**Metareview:**

While the paper's approach is applicable only to MDPs with a known relational model expressible in the PPDDL language, the reviewers unanimously find it to be an interesting, novel advance for this class of settings.

**Award:**

No

---

### Decision · Program_Chairs · 2022-09-14

Accept